# Limited production of sulfate and nitrate on front-associated dust storm particles moving from desert to distant populated areas in northwestern China

Feng Wu[1], Daizhou Zhang[2], Junji Cao[1,3], Xiao Guo[1,4], Yao Xia[1,4], Ting Zhang [1], Hui Lu[5], Yan Cheng[3]

5   [1] Key Laboratory of Aerosol Chemistry & Physics, and State Key Laboratory of Loess and Quaternary Geology, Institute of Earth Environment, Chinese Academy of Science, Xi'an, China
[2] Faculty of Environmental and Symbiotic Sciences, Prefectural University of Kumamoto, Kumamoto, Japan
[3] Institute of Global Environmental Change, Xi'an Jiaotong University, Xi'an, China
[4] School of Tropical Eco-Environment Protection, Hainan Tropical Marine University, Sanya, China
10   [5] Institute of Desert Meteorology, China Meteorological Administration, Urumqi, China

*Correspondence to*: Feng Wu (kurt_wf@ieecas.cn); Daizhou Zhang (dzzhang@pu-kumamoto.ac.jp)

**Abstract.** Sulfate and nitrate compounds can greatly increase the hygroscopicity of mineral particles in the atmosphere and, consequently, alter the particles' physical and chemical properties. Their uptake on long-distance transported Asian dust particles within mainland China has been reported to be substantial in previous studies, but the production was very inefficient in other studies. We compared these two salts in particles collected from a synoptic scale, mid-latitude cyclone-induced dust storm plume at the Tengger desert ($38.79^\circ$ N, $105.38^\circ$ E) and in particles collected in a postfrontal dust plume at an urban site in Xi'an ($34.22^\circ$ N, $108.87^\circ$ E) when a front-associated dust storm from the Tengger desert arrived there approximately 700 km downwind. Results showed that sulfate concentration was not considerably different at the two sites while nitrate concentration was slightly larger at the urban site than that at the desert site. The estimated nitrate production rate was 4-5 ng per μg mineral dust per day, which was much less than that in polluted urban air. The adiabatic process of the dust-loading air was suggested to be the reason for the absence of sulfate formation, and the uptake of background $HNO_3$ to be the reason for the small nitrate production. To the extent of our investigation of published literature, the significant sulfate and nitrate in dust storm-associated samples within the continental atmosphere reported in previous studies cannot be confirmed to be really produced on desert dust particles and the contribution from locally-emitted and urban mineral particles or from soil-derived sulfate was likely substantial, because the weather conditions in those studies indicated that the collection of the samples was started before dust arrival or the air from which the samples were collected was a mixture of desert dust and locally-emitted mineral particles. These results suggest that the production of nitrate and sulfate on dust particles following cold fronts is likely limited when the particles move from the desert to populated areas within the continent. For an accurate quantification of sulfate and nitrate formed on long-distance transported desert dust particles at downwind populated areas in eastern China, efforts are indispensable in dust collection to minimize any possible influence by locally-emitted particles or, at least, to ensure that the samples are collected after dust arrival.

**Keywords**: Asian dust, Tengger desert, long-distance transport, chemical conversion, background $HNO_3$

# 1 Introduction

Mineral dust particles constitute a substantial fraction of atmospheric aerosol mass and plays various roles in atmospheric physics and chemistry (Bi et al., 2011; Dentener et al., 1996; Fu et al., 2009; Sokolik and Toon, 1996; Tegen et al., 1996). Dust particles at their source areas are mainly composed of quartz, clays, micas, feldspars, carbonates (primarily calcite, $CaCO_3$), and other minor minerals (Usher et al., 2003). While suspended, they may be altered by the uptake of gases and smaller particles and by surface reactions. Laboratory studies have demonstrated the formation of sulfate and nitrate on dust particles upon exposure to reactive gases such as $NO_x$, $HNO_3$, $NO_3$, $N_2O_5$ and $SO_2$ (Usher et al., 2003). The formation of salts on the particles can enhance the solubility of the particles, lower their effective deliquescence relative humidity (RH), and alter their size and physical state in association with atmospheric conditions (Semeniuk et al., 2007). These changes in turn feed back into the activities of dust particles in various chemical and physical processes in the atmosphere (Bauer and Koch, 2005), such as the enhancement of bioavailable iron (Meskhidze, 2003) and the removal of acidic gases in the atmosphere (Dentener et al., 1996; Zhang and Carmichael, 1999).

Field studies have shown different results regarding the formation of sulfate and nitrate on dust particles and some results are contradictory. Many studies reported substantial sulfate and nitrate on the surface of Asian dust particles after the particles were transported over long distances in the atmosphere (Cao et al., 2003; Huang et al., 2010; Li and Shao, 2009; Mori, 2003; Nie et al., 2012; Nishikawa et al., 1991; Qi et al., 2006; Ro et al., 2005; Sun et al., 2010; 2004; Wang et al., 2005b; 2007; Wu and Okada, 1994; Zhao et al., 2011). In contrast, an early study of Zhang and Iwasaka (1999) found that sulfate and nitrate were rarely formed on Asian dust particles which had been transported over a long distance in inland China after about two days. A study at the Taklimakan desert pointed out that in some cases the content of sulfate in dust particles might not change even when the particles traveled over a long distance. The dust particles contained substantial sulfate (~4% by mass) which was from the surface soil (Wu et al., 2012). It was also found that, for a dust plume lofted from the surface by a synoptic mid-latitude cyclone, the plume did not mix significantly with adjacent air parcels polluted by anthropogenic sources; the dust plume and the polluted air were separated as two air parcels by the cold front associated with the cyclone (Bates et al., 2004; Tsai et al., 2014; Wang et al., 2013; Zhang et al., 2005). Some measurements of chemical composition of long-distance transported dust particles have also shown that most of the dust particles were not altered chemically and were externally mixed with species produced in the air via gas-to-particle reactions, such as sulfate and nitrate (Denjean et al., 2015; Song et al., 2005). These results leave a question: why are there so much different rates of sulfate and nitrate production during dust transport to polluted areas?

In April 2014, we collected a series of atmospheric particle samples during a cyclone-induced dust storm at the eastern edge of the Tengger desert, which is one of the most significant sources of Asian dust (Wang et al., 2012; Zhang et al., 2003). We also collected a series of samples at Xi'an, a large city in northwestern China when a dust storm from the Tengger desert passed there after travelling about 6 hours following a cold front. In this study, we compare the concentrations and mass fractions of sulfate and nitrate in the samples at the two sites, examine the production of nitrate and sulfate on desert dust

particles after the particles were transported from the desert to the populated area, and to understand the chemistry/aging on dust.

## 2 Particle Collection and Analysis

The observation site at the Tengger desert was located at an active sand dune at a location called Tonggunao'er along the northeastern rim of the desert (38.79ºN, 105.38ºE; Fig. 1). The closest village, with a population less than 200, is about 5 km to the east of the site and the nearest city is Bayan Hot (Inner Mongolia Autonomous Region, China) about 35 km to the east of the site (Fig. S1). Anthropogenic pollutants from the village and the city may arrive at the site if the wind direction is from the east. Backward trajectories of air masses (Fig. S2) and the simulation (Fig. S3a) of the on-line Chemical Weather Forecasting System (CFORS, developed and open on-line by NIES and Kyushu University, Japan: http://www-cfors.nies.go.jp/~cfors/index-j.html) showed that a dust storm was induced by a synoptic scale, mid-latitude cyclone in the southwest part of the Mongolia on April 23, 2014. The resulting dust plume was then transported southeastward and passed the sampling site on the morning of April 24 (Fig. S3a).

Observations were carried out between 06:30 BST (Beijing standard time: GMT + 08:00) and 15:00 BST on April 24, 2014. Particles were collected using a homemade filter pack sampling system, which consisted of one Teflon front filter for collecting particulate matter and one back filter for the collection of gas-phase species. The flow rate of 16.7 litres per minute was controlled with a mass flow controller (SmartTrak 50, Sierra). The filters were changed every 2 hours, and the collection of the fourth sample was stopped when it started to snow (Table 1). Field blank filters were prepared and obtained by mounting filters in a sampling system in a similar way to the particle collection for 2 hours without pumping air. Right after the sample collection of each filter, the filter was put into a polystyrene petri dish, which was in turn sealed in a plastic bag and stored into refrigerators at -1°C until subsequent analyses. Meteorological conditions including surface pressure, temperature, relative humidity, wind speed, and wind direction were monitored with a weather tracker (Kestrel 4500, Kestrelmeters). The cold front passed the site between 04:00 and 04:30 BST on 24 April, which was characterized by the rapid decrease of relative humidity, the sudden change of wind direction from south to north, the growth of wind speed, and the gradual increase of pressure (Fig. 2a). Therefore, all samples were collected after the passage of the cold front. This sample collection ensured that mineral particles collected on the filters were dominated by dust particles from the desert, and possible influence of anthropogenic pollutants from the village or the city was suppressed.

Xi'an (34.22ºN, 108.87ºE) lies in central China, approximately 700 km from the Tengger desert (Fig. 1). The observation in Xi'an was carried out on the roof of a building (10 m above ground) on the campus of the Institute of Earth Environment. The institute is located in the southwest area of the city, and its surroundings are residences, streets and office buildings. There are no large, continuous sources of anthropogenic pollutants such as factories or agriculture fields near the institute. Previous studies at this site have revealed that the local pollutants are mainly from traffic and the particulate pollutants

mainly include particles from engines of vehicles, road dust, and construction dust, all of which have been characterized by contents of crustal elements, sulfate, organic matter, nitrate, and ammonium (Huang et al., 2014a; Zhang et al., 2011).

Particles were collected on May 1, 2014 when dust-loading air passed Xi'an. Backward trajectories of air masses (Fig. S2) and the CFORS simulation (Fig. S3b) showed that, similar to the dust storm observed at the Tengger desert, this dust storm was also induced by a cyclone in the southwest part of the Mongolia and moved southeastward (Fig. S3b). It passed the site of the Tengger desert on the evening of April 30 and arrived at Xi'an on the morning of May 1. Particle collection was carried out between 07:00 BST and 19:00 BST on May 1. The same sampling system and the same type filters as those used for the particle collection at the desert site were used. Samples were collected at a time interval of 1 or 2 hours. In total, 7 samples were obtained from the dust-storm approach to its dissipation (Table 1). Each sample filter was put into a polystyrene petri dish, which was in turn sealed in a plastic bag and stored in a refrigerator at -1°C until subsequent analyses. Meteorological conditions were monitored by the Kestrel 4500 weather tracker. The sudden decrease of relative humidity showed that the arrival of the cold front of the cyclone occurred between 9:30 and 11:30 BST (Fig. 2b). Therefore, the first sample was collected before the arrival of the cold front, the second and third samples were collected in the frontal air and the fourth, fifth, and sixth samples were collected after the passage of the cold front. Unfortunately, the fourth sample was not available for analysis because the collection system was blown down by wind when this sample was collected. The concentrations of $SO_2$ and $NO_2$ were measured by an UV fluorescence analyzer (Ecotech, model EC9850) and a chemiluminescence analyzer (Thermo, model 42i) and were recorded in units of ppb at a time interval of 5 minutes. The lowest detection limit was approximately 0.04 ppb for $SO_2$ and 0.4 ppb for $NO_2$. Since the dust occurring in Xi'an had the source same as and a transport route similar to that at the desert site (Fig.S2, Fig.S3), the comparisons between the samples collected at the desert site and the Xi'an site can show changes of dust particles during the transport although the samples were not from the same dust event.

Teflon-membrane filter samples were equilibrated in a temperature and relative humidity controlled environment (20–23°C and 35–45% RH, respectively) before gravimetric analysis to minimize particle volatilization and aerosol liquid water interferences. Filters were weighed before and after sampling using a ME 5-F electronic microbalance (Sartorius, GOTTINGEN, Germany) with a sensitivity of ± 0.001 mg. The precision of multiple weighings for unexposed and exposed filters was smaller than ± 0.010 and ± 0.015 mg, respectively. Filters were exposed to a low-level radioactive source (500 pCi of polonium-210) before sample weighing to remove static charge. Mass concentrations were calculated with the difference of weight before and after sampling and the volume of sampled air. Half of each filter was analyzed to quantify water-soluble components in particles. Each sampled filter was initially wetted with 200 µl ethanol. Water-soluble components were extracted by ultrasonic agitation in 10 mL distilled water. The extraction solution was filtered with 0.45 mm pore size microporous membranes, and then the filtrate solution was stored at 4°C until subsequent analyses. An ion chromatograph (Dionex DX-600) was used to quantify sulfate ($SO_4^{2-}$), nitrate ($NO_3^-$), chloride ($Cl^-$), sodium ($Na^+$), potassium ($K^+$), ammonium ($NH_4^+$), calcium ($Ca^{2+}$), and magnesium ($Mg^{2+}$) in the solution. Calibration curves were constructed from the peak areas of the chromatograms which were produced from a series of mixed standards. Samples and blanks collected at both sites were

analyzed with replicates and surrogates following standard lab protocols. The relative uncertainty in the mass percentage of sulfate and nitrate, according to the surrogates, was less than 10%.

An energy dispersive X-Ray fluorescence (ED-XRF) spectrometry (Epsilon 5 ED-XRF, PANalytical B. V., the Netherlands) was used to quantify elements in the samples of the remained parts of sample filters. Five crustal elements (K, Ca, Ti, Mn, Fe and Ba) and two common anthropogenic trace elements (Zn and Pb) were quantitatively determined. Analytical uncertainties, as checked by parallel analysis of the NIST standard reference material (SRM-2683), were about or less than 10% for the detected elements.

## 3 Results and discussion

### 3.1 Sulfate

In the dust plume at the Tengger desert, the concentration of sulfate ranged from 39 to 59 $\mu$g m$^{-3}$. The relative amount of sulfate in the dust samples, i.e. the mass fraction of sulfate in the samples, was between 1.1% and 1.2% and the average was 1.2% (Table 2). These values were close to the levels of the relative mass ratios of sulfate in TSP or PM$_{10}$ in samples collected under dust conditions at the Gobi Desert which were reported in previous studies (Table S1).

The concentration of SO$_4^{2-}$ at Xi'an varied in a large range as the front of the dust-loading cyclone was approaching, passing and leaving (Fig. 3). It was 17 $\mu$g m$^{-3}$, contributing 4% of aerosol mass, in the prefrontal air, i.e. before the dust arrival. As the front was passing, the concentration decreased rapidly. In the postfrontal air, the concentration was 3.8 $\mu$g m$^{-3}$ right after the front passage, 3.5 $\mu$g m$^{-3}$ two hours after the passage, 3.4 $\mu$g m$^{-3}$ four hours after the passage. The average in the postfrontal air was 3.5 $\mu$g m$^{-3}$. The relative amounts of sulfate in these samples were 0.9%, 1.1%, and 1.8%, respectively, and the average was about 1.3 %. In comparison with that in the prefrontal air, the concentration in the postfrontal air was very small and approximately constant, although the relative amount increased gradually as the front left.

In cyclones moving eastward across northern China, the source and consequently, the compositions of major particles in the air before and after cold fronts are different although they are in the same cyclones (Hu et al., 2016; Niu et al., 2011). Prefrontal air usually moves slowly from the south or southwest directions toward the north or northeast, and is usually warm and humid. Particles in prefrontal air are originated mainly from local or regional areas and they are usually dominated by anthropogenic pollutants (Li et al., 2012). The postfrontal air moves more rapidly from north or northwest direction and is usually cold and dry. Particles in postfrontal air are those lifted by cold fronts of the cyclones at the places they pass, and long-distance transported dust particles are usually the majority if the cyclones have caused dust storms at the arid and semi-arid areas in northwestern China (Wang et al., 2005a). The cold fronts are the boundaries between the local or regional anthropogenically-polluted air and the long-distance transported air because the movement of air on a synoptic scale is approximately adiabatic, i.e. the air is hardly mixed with thermodynamically-different air it meets, although some small-scale mixing might occur in the front. Aerosol particles at the time period of front passage should be dominated by both locally

originated and long-distance transported ones. The rapid decrease of sulfate with the passage of the cold front at Xi'an was consistent with the increase of long-distance transported dust.

$NH_4^+$ is one of the major water-soluble species in aerosol particles, and can be remarkably enhanced by anthropogenic emissions. We found that its concentration was close to the lowest detection limit in dust samples at the desert site. This fact makes $NH_4^+$ a good indicator for examining the influence of local or regional anthropogenic particles on the samples observed at Xi'an. The variation of $NH_4^+$ during the sampling period is also shown in Fig. 3. With the increase of long-distance transported dust particles, $NH_4^+$ decreased remarkably as the front was passing the site, and was very low in the frontal air. In the postfrontal air, $NH_4^+$ concentration was lower than the detection limit in the first sample and increased slightly in the second and third samples. These results indicate that the composition of particles in the postfrontal air was close to the state of dust particles at the desert areas, whereas the composition was gradually affected by local emissions afterwards. Zn and Pb are two common anthropogenic trace elements in urban air. Their ratios to Fe in the dust samples in the postfrontal air were much lower than those in the prefrontal air and very close to those in the desert air (Table 3), further suggesting the limited influence of pollution on desert dust particles in the postfrontal air.

The relative amount of $SO_4^{2-}$ in dust samples at the desert, 1.1 to 1.2% in mass, was similar to or even larger than the relative amount at Xi'an (the relative amount in the first sample in the postfrontal air was 0.9%). This result indicates that sulfate was rarely produced on dust particles during the particles travelled from the desert to the distant urban area. Heterogeneous reactions involving $SO_2$ on mineral particles was the major processes for sulfate production on the particles. The conversion of $SO_2$ to sulfate by heterogeneous reactions on particles is much more efficient under humid conditions than under dry conditions (Usher et al., 2003). The relative humidity was less than 40% during the dust-storm episode in Xi'an (Fig. 2b), which did not favor the formation of sulfate on the surface of mineral components (Huang et al., 2014b). Moreover, the cold air lofting the dust-storm particles was from arid or semi-arid areas in the southwest part of the Mongolia, where $SO_2$ emission is usually weak (Fig. S4). The concentration of $SO_2$ in the postfrontal air was nearly close to the detection limit (Fig. S5), which was much smaller than the concentration in the prefrontal air (~ 20 ppb). Therefore, sulfate was hardly formed on the dust particles due to the lack of $SO_2$ and the dry condition. The postfrontal air had passed some populated areas between the desert and Xi'an, where anthropogenic $SO_2$ emission was usually observed due to human activities (Wang et al., 2011). However, the postfrontal air did not pick up any accumulated air pollutants on the way. Anthropogenic pollutants that might have been taken into the air were those freshly emitted at the moment of the air passage. Such pollutants should not have a considerable influence on the dust. Otherwise, (1) the movement of the dust-loading air should not have been an adiabatic process, the reason for the cold front occurrence when arriving at Xi'an; (2) the front should have disappeared; (3) some $NH_4^+$ should have been present; and (4) sulfate content in the samples at Xi'an should be larger than the desert-sample level. The vertical thermodynamic structure near the surface at the two sites became more stable when dust occurred (Fig.S6: from the homepage of Atmospheric Soundings of the University of Wyoming, http://weather.uwyo.edu/upperair/sounding.html), indicating that the dust plume layer established at the dust source was not mixed with air of different chemical (gas and particulate phase) composition from above during the advection.

There was a small amount of sulfate in the dust samples at Xi'an. The concentration was much smaller than that in the dust samples at the desert area. However, the relative level of sulfate in total aerosol mass in the postfrontal air samples (0.91-1.75%) was close to and even smaller than that at the desert area (1.19% on average). This result means that the sulfate in the dust samples at Xi'an was very likely one of the original components of the dust particles, i.e., the so-called soil-derived sulfate in desert dust. It has been found that dust at desert areas contained substantial soil-derived sulfate (Abuduwaili et al., 2008; Sun et al., 2010; Wang et al., 2012; Wu et al., 2012; Yabuki et al., 2005; Zhang et al., 2009) and the sulfate was confirmed in long-distance transported dust in the downstream areas in a small number of studies (Wang et al., 2014; 2016b; 2007; Wu et al., 2012). For these reasons, we consider that the sulfate detected in the dust samples right after the cold front was mainly from the desert areas as soil-derived one, rather than the sulfate produced by chemical conversions on the particle surface when the particles floated in the air.

**3.2 Nitrate**

At the desert site, $NO_3^-$ concentration in dust samples was 4-6 $\mu g\ m^{-3}$ and the average was 5 $\mu g\ m^{-3}$. The relative amount of $NO_3^-$ ranged between 0.11% and 0.12%, and the average was 0.12%. At Xi'an site, similar to $SO_4^{2-}$, the concentration of $NO_3^-$ varied in a large range as the dust-loading cyclone passed, with the concentration high in the prefrontal air and low in the postfrontal air. Right after the passage of the cold front (the first sample in the postfrontal air), the concentration of $NO_3^-$ was 0.9 $\mu g\ m^{-3}$ and it occupied 0.2% of the aerosol mass. These values were close to the levels of the relative mass ratios of nitrate in TSP or $PM_{10}$ in samples collected under dust conditions at the Gobi Desert which were reported in previous studies (Table S1).

The relative amount in this sample was about twice of that in the desert samples although it was the lowest in the samples at Xi'an site, indicating that nitrate was likely produced on dust particles during their travel to Xi'an. We assume that the removal of dust particles from the dust plume was independent from the chemical components of dust particles. This assumption is reasonable because the dust plume was relatively dry during its movement from the desert area to the urban area and the settling of dust particles under such conditions depends on particles' size only (Zhang, 2008). The concentration of nitrate in dust samples at the desert site was used as the referential value of the part of nitrate originating from the desert areas in the samples at Xi'an areas. The results show that nitrate production on the dust particles was 1.0-1.1 ng per $\mu g$ dust (approximately 0.1-0.11% in mass) during the dust movement from the desert to Xi'an. Since the dust plume took approximately six hours to move from the desert to Xi'an, this increase of nitrate was equivalent to a production rate of 4-5 ng nitrate per $\mu g$ dust per day. Note this rate should be the maximum rate because not all the nitrate must have been produced on dust particles and the increase of the relative amount of nitrate during the movement of a dust plume from the desert to Xi'an could have been the consequence of possible difference of removal rates of dust particles and nitrate-containing particles. Anyway, the estimated rate is much smaller than that in polluted urban air, where secondary nitrate usually accounts for 2-6% of aerosol loading (Wang et al., 2003) and the production rate is 20~60 ng nitrate per $\mu g$ aerosol per day, if we consider the residence time of particles in polluted air is 24 hours.

In general, nitrate on dust particles is produced on the surface via heterogeneous conversions, or the uptake of $HNO_3$ which is formed via homogeneous reactions in the atmosphere. Model studies have shown that the latter one is the major route for nitrate formation on dust particles, and the contribution of the former route, in particular under dry conditions, is very small (Fairlie et al., 2010; Song and Carmichael, 2001). Desert soil hosts the premier natural nitrate minerals on the earth (Walvoord, 2003). Nitrate in desert soil can be reduced to $NO_x$ through microbiological denitrification (Hartley and Schlesinger, 2000) and abiotic thermal decomposition (McCalley and Sparks, 2009). A background-like nitrate, which is about 2-8 μg m$^{-3}$ and supposed to be in the form of nitric acid, has been found in desert air (Wu et al., 2014). Such $HNO_3$ could be absorbed and transformed into nitrate on dust particles during the dust movement from the source region to Xi'an.

A first-order chemical kinetic equation was used to estimate the uptake of $HNO_3$ during the transport.

$$c_t = c_0 \, e^{-kt}, \tag{1}$$

where $c_t$ is the concentration of $HNO_3$ at transport time $t$, $c_0$ is the initial concentration of $HNO_3$ in the air mass, and $k$ is the first order rate at which the gaseous precursor is taken up by dust. The reaction rate can be calculated as below:

$$k = \frac{1}{4} v_{HNO_3} \gamma_{HNO_3} A_P, \tag{2}$$

where $v_{HNO_3}$ is the mean molecular speed of $HNO_3$ and is taken as $3.0 \times 10^4$ cm s$^{-1}$ (Fairlie et al., 2010). $\gamma_{HNO_3}$ is the reactive uptake coefficient for $HNO_3$ on dust particles. Measurements in laboratory experiments showed the uptake coefficient of $HNO_3$ on mineral dust ranged from $5.2 \times 10^{-5}$ (Underwood et al., 2001) to $5 \times 10^{-3}$ (Song et al., 2007). The coefficient was a function of relative humidity (Vlasenko et al., 2006), and estimated to be $\sim 2 \times 10^{-4}$ under experimental conditions of 30~40% relative humidity (Fairlie et al., 2010). $A_P$ (cm$^2$ cm$^{-3}$) is the total surface area of dust particles and is determined by the loading of dust particles and its specific surface area. Surface area analysis has shown the specific area of Gobi dust to be approximately 110 cm$^2$ mg$^{-1}$ (Underwood et al., 2001). The dust load by mass (in μg m$^{-3}$) initially at the Tengger desert for the dust plume observed at Xi'an is estimated to be 1187 μg m$^{-3}$, based on the dust load of the first sample of postfrontal air (415.4 μg m$^{-3}$) and the relationship between dust concentration and its transport distance suggested by Mori et al. (2002).

The reaction rate for $HNO_3$ gas uptake on mineral dust was $1.96 \times 10^{-4}$ s$^{-1}$. The concentration of $HNO_3$ after 6 hours reduced to ~1% of the initial concentration, indicating that almost all $HNO_3$ could be transformed into nitrate on the dust particles during their transport. As described above, nitrate produced on the dust samples was 0.1-0.11%, indicating the concentration of $NO_3^-$ increased by 1.2-1.3 μg m$^{-3}$. To produce this amount of nitrate, the $HNO_3$ concentration in the dust-loading air should be approximately 1.2-1.3 μg m$^{-3}$ on average. Unfortunately, there are no data on nitric acid in the air over the Tengger desert for further comparison. The concentration of nitrate (including particulate and gaseous phases) at Taklimakan desert in northwestern China was 2~8 μg m$^{-3}$ (Wu et al., 2014), which was in the same range as found for the nitrate concentration in the dust at our Xi'an site.

**3.3 Inter-comparisons and implication**

There have been studies on dust-associated sulfate and nitrate in aerosol particles downwind of areas in mainland China (in Table 4). Zhao et al. (2007) investigated the evolution of air pollutants when an extremely strong dust storm from Gobi Desert near China-Mongolia border passed Beijing and confirmed the rapid decrease of sulfate and nitrate after the passage of the dust-associated cold front. The relative amounts of sulfate and nitrate in the postfrontal air were 0.77% and 0.08%, respectively, which were close to the relative amounts of the salts we observed at the desert site and Xi'an site in this study. Wang et al. (2014) reported hourly sulfate and nitrate in aerosols at Xi'an during a dust storm period on March 9, 2013. The dust storm originated also from the Gobi Desert in the southwest part of the Mongolia, similar to the dust cases in this study. After the passage of the cold front, the relative amounts of sulfate and nitrate were 1.05% and 0.21%, respectively, which were also very close to the results of this study.

A number of studies in China reported that dust particles significantly enhanced the formation of sulfate and nitrate when dust plumes advected over urban areas (Li and Shao, 2009; Li et al., 2014; Nie et al., 2012; Qi et al., 2006; Sheng et al., 2003; Wang et al., 2013; Zhao et al., 2011; 2007), which are very different from the conclusions of this study. We carefully examined the available meteorological records for the dust episodes in those studies. In cases of cold-front associated dust (Li et al., 2014; Qi et al., 2006; Sheng et al., 2003; Zhao et al., 2007), the samples used in those studies were collected repeatedly in pre-fixed time periods without a careful consideration of the time of dust arrival. That means the samples were not separately collected from defined prefrontal air and postfrontal air masses and some samples used for analysis included particles from both prefrontal air and postfrontal air. The results from such samples would show the presence of substantial sulfate and nitrate. However, the sulfate and nitrate must have been contributed by particles in prefrontal air, which should be from local or regional areas and abundant in sulfate and nitrate. A recent publication about the passage of a dust storm passing Beijing with high-time resolution, on-line records demonstrated clearly the separation of the prefrontal pollutants and the postfrontal dust plume (Hu et al., 2016), further indicating the necessity of separating particles in prefrontal air and postfrontal air for an accurate description of salt origin in dust samples.

In addition, mineral ions or crustal elements (e.g. $Ca^{2+}$ ion, or elemental Al) have been frequently applied as the indicators of mineral dust in studies (Nie et al., 2012; Zhao et al., 2007). Samples with the highest loading of mass or crustal compositions (e.g. $Ca^{2+}$, Al) were frequently regarded as the samples of long-distance transported dust. However, the samples in those studies were actually collected from the front air as we described above and were a mixture of long-distance transported dust particles and locally- and regionally-originated aerosols. For example, in the study of Zhao et al. (2007), the mineral/TSP (total suspended particulate matters) ratios in samples of the highest TSP loading were significantly lower than those in samples collected after the occurrence of maximum aerosol loading, indicating that the samples at the highest TSP moment were not dust particles from desert areas only. The indicators of mineral crustal ions or elements (e.g. $Ca^{2+}$ ion, or elemental Al) may actually cause large uncertainties in the explanation of the origins and sometimes even misunderstandings on the origins of aerosol particles. The reason is that, besides road dust and construction dust, coal burning is a major source

of mineral components in aerosols in China, and emits particulate matter that is abundant in mineral elements such as Si, Al, Ca and Fe (Chen et al., 2012). Coal burning emissions have been proven to cause significant air pollution in China (Cao et al., 2005; Wang et al., 2015; 2016a). Anthropogenic pollutants are usually present in prefrontal air. If only the presence of substantial mineral elements such as Ca, Al or Si is used as the indicator of the occurrence of mineral dust particles from desert areas, anthropogenic pollutants such as road dust and particles emitted from coal burning would have been categorized as desert dust particles. Such an examination would lead to a result of the occurrence of substantial sulfate- and nitrate-containing dust particles in samples.

In many cases, dust was observed in cyclonic disturbances with weak fronts or without fronts (Cao et al., 2003; Nie et al., 2012; Wang et al., 2013). Cold fronts could not be confirmed clearly at places of dust arrival in those studies. The cases usually occur in the south parts of China after the cold and dry air from the north lose their adiabatic state, and the postfrontal air arriving at such places has, to an extent, mixed with the local and regional air. Samples of aerosol particles after dust arrival at such places contain both long-distance transported dust particles and locally or regionally emitted pollutants. For example, Wang et al. (2013) reported the occurrence of nitrate and sulfate in particles during two extreme dust storms in Shanghai, a megacity in eastern China, from 20 to 22 March 2010 and from 26 to 27 April 2010. Weather charts (see Fig. 8 in the paper) showed the consecutive transport of anthropogenic air masses and dust storm plumes to Shanghai during the dust periods either with a cold front arrival (the previous case; equivalent to dust case with front in this study) or by the stimulation of a cold front event even though the front did not extend to Shanghai (the latter case; dust case without front). The relative amounts of sulfate and nitrate in samples at Shanghai during this period were 2.7% and 1.4% in the previous case and 9% and 5.9% in the latter case. Anthropogenic sulfate and nitrate in particles from the local and regional areas in the latter case would have appeared in the samples although they were not produced on dust particles from desert areas.

To further examine the situation of previously-reported sulfate and nitrate formation on dust-storm particles at populated areas in eastern China, we investigated all published papers we found on this subject, and carefully checked the records of sample collection and the available meteorological conditions when the samples were collected in those studies, from the papers and officially public web sites for historical meteorological records. The papers were separated into three groups according to the records and meteorological conditions, as mentioned above. The first group includes papers in which the sample collection records were vague, and we are unable to make clear if the samples were dominated by desert dust particles or contained a large fraction of locally-emitted particles (Table S2a). It is not sure that the sulfate and nitrate reported in those papers were really on desert dust particles or not. The second group includes papers in which the sample collection was started before the arrival of the fronts of dust-loading cyclones or the fronts had disappeared and front-associated dust-loading air had mixed with locally-polluted air (Table S2b). That means the samples in the studies contained not only long-distance transported desert dust particles but also locally-emitted mineral particles, such as road dust, construction dust and fly ashes. In such samples, the sulfate and nitrate must have been substantially contributed by locally-emitted mineral particles, as we discussed above. The third group includes papers in which results from samples of locally- or regionally-originated particles in prefrontal air and from samples of long-distance transported desert dust particles in postfrontal air can be identified

(Table S2c). The production of sulfate and nitrate in the postfrontal dust samples was all very limited, and the production in prefrontal samples was significant, which is consistent with our results in this study.

**4 Conclusion**

Dust particles were collected at the Tengger desert and Xi'an during two dust storm periods. Meteorological records showed these two dust storms originated from the same source region and had similar transport routes. The comparison of sulfate and nitrate of dust aerosol at the two places indicated that the production of sulfate and nitrate on dust particles following cold fronts was limited when the dust moved from the desert to the populated area. The adiabatic process of the dust-loading air movement was most likely the reason for the absence of sulfate formation, and the uptake of $HNO_3$ was for the small nitrate production.

Significant sulfate and nitrate in dust storm periods in China reported in previous studies in reality for the most part probably did not link to reactions on the dust surface. They were likely from locally-emitted and urban mineral particles, in addition to soil-derived sulfate. The major reason is that, in those studies, the air from which the samples were collected had been significantly influenced by local emissions. Without a proper evaluation of the contribution of sulfate and nitrate in the samples by locally-emitted and urban mineral particles, i.e., non-desert mineral particles, it is not safe to attribute all the detected sulfate and nitrate to the production on dust-storm particles.

The results of this study are from the comparison of dust particles in two dust storms: one at the dust source area, and another at an urban area after long-distance transport. Although the thermodynamic structure of the dust-loading air in the two cases was similar and comparative, data from multiple cases of same dust storms at desert areas and downwind populated areas are needed to make the present conclusions more accurate and confident. In addition, the conclusions were derived for front-associated dust storm particles. The adiabatic nature of the postfrontal air during its long-distance movement kept the air dry and hardly polluted by accumulated anthropogenic pollutants from areas where it passed. There are other types of airborne dust particles in China, such as floating dust. The movement of the air loading floating dust is usually slow and not adiabatic, and the air is usually well mixed with locally-emitted pollutants, which is very different from postfrontal air. It can be expected that floating dust particles could be more frequently changed to sulfate- and nitrate-carriers via surface chemical reactions than front-associated dust storm particles in urban areas. However, how to separate the sulfate and nitrate produced on floating dust particles from those produced on locally-emitted mineral particles is still a big challenge in field observations, because floating dust particles and locally-emitted mineral particles coexist in urban air when floating dust occurs.

**Author contribution**: Feng Wu and Daizhou Zhang designed the experiments; Xiao Guo, Yao Xia and Ting Zhang collected the samples and analyzed them. Juji Cao, Yan Cheng and Hui Lu reviewed and commented on the paper; Feng Wu and Daizhou Zhang prepared the manuscript with contributions from all co-authors.

**Competing interests**: The authors declare that they have no conflict of interest.

**Acknowledgments**

This work was supported by the "Strategic Priority Research Program" of the Chinese Academy of Sciences (Grant No. XDB05000000), the NNSF of China (Grant No. 40872211). It was also partly funded by the Grant-in-Aid for Scientific Research (B) of JSPS (No.16H02942). The authors thank Prof. Jay Melton and Prof. Morrow Stewart for their reviewing of grammar and language. We also thank Dr. Shuyu Zhao for her providing the figure of $SO_2$ emission distributions.

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

Table 1: Summary of weather conditions at the sample collection

| Samples ID | Sampling time (BST[c]) | Pressure (hPa) | Temperature (°C) | RH (%) | Wind Direction | Wind Speed (m s$^{-1}$) |
|---|---|---|---|---|---|---|
| Tengger Desert (April 24, 2014) | | | | | | |
| T1 | 06:32-08:32 | 870.2-872.5 | 9.5-11.0 | 30-35 | NW | 5.7-13.6 |
| T2 | 08:42-10:42 | 872.4-874.7 | 8.0-9.4 | 31-37 | NW | 5.6-11.4 |
| T3 | 10:51-12:51 | 875.2-876.7 | 5.5-7.7 | 31-39 | NW | 6.0-14.6 |
| T4[a] | 13:02-15:03 | 876.2-878.1 | -1.4-4.8 | 40-97 | NW | 5.8-12.3 |
| Xi'an (May 1, 2014) | | | | | | |
| X1 | 07:16-09:16 | 965.6-968.1 | 17.6-19.6 | 63-72 | W | 0-1.8 |
| X2 | 09:20-10:20 | 968.2-969.7 | 19.5-21.7 | 45-67 | NW | 0-2.5 |
| X3 | 10:22-11:22 | 970-971.7 | 21.2-22.2 | 39-50 | NW | 0.5-3.2 |
| X4[b] | 11:27-12:27 | 971.7-972.7 | 20.1-21.3 | 40-43 | NW | 0.6-3.4 |
| X5 | 12:28-14:28 | 972.7-973.5 | 20.0-20.7 | 38-42 | NW | 0.4-2.9 |
| X6 | 14:38-16:38 | 972.4-973.2 | 19.0-21.6 | 38-47 | NW | 0-5.4 |
| X7 | 16:43-19:43 | 972.5-973.1 | 18-21.8 | 38-48 | NW | 0-3.0 |

[a] Not suitable for comparison because of snow and results from this sample were excluded for further analysis.
[b] Not available for analysis because the collection system was blown down by wind.
[c] Beijing standard time (8 hours prior to GMT).

Table 2: Concentration (Conc., in μg m$^{-3}$) of TSP ([M]), sulfate ([$SO_4^{2-}$]), nitrate ([$NO_3^-$]) and ammonia ([$NH_4^+$]) at the desert site in the dust episode. Also included are the relative amount (R. M., in %) of the three ions in TSP

| Samples | [M] | [$SO_4^{2-}$] | | [$NO_3^-$] | | [$NH_4^+$] | |
|---------|-----|------|------|------|------|------|------|
| | | Conc. | R. M. | Conc. | R. M. | Conc. | R. M. |
| T1 | 4754 | 59 | 1.2 | 5.9 | 0.12 | 0.27 | <0.01 |
| T2 | 4487 | 54 | 1.2 | 5.1 | 0.11 | 0.22 | <0.01 |
| T3 | 3481 | 39 | 1.1 | 3.8 | 0.11 | ND$^a$ | ND |
| Ave. | 4241±672 | 51±10 | 1.2±0.1 | 5.0±1.1 | 0.12±0.11 | 0.16±0.14 | <0.01 |

$^a$ Not detected

Table 3: Mass ratios of Ca, Fe, Ti, Mn, Ba, Zn and Pb to Fe in the samples at the two sites

| Samples | Ca/Fe | K/Fe | Ti/Fe | Mn/Fe | Ba/Fe | Zn/Fe | Pb/Fe |
|---|---|---|---|---|---|---|---|
| Tengger Desert (April 24, 2014) | | | | | | | |
| T1 | 1.47 | 0.54 | 0.084 | 0.023 | 0.013 | 0.003 | 0.0014 |
| T2 | 1.47 | 0.55 | 0.082 | 0.023 | 0.013 | 0.0023 | 0.0011 |
| T3 | 1.57 | 0.57 | 0.086 | 0.024 | 0.012 | 0.002 | 0.0009 |
| Xi'an (May 1, 2014) | | | | | | | |
| X1[a] | NA | NA | NA | NA | NA | NA | NA |
| X2 | 1.86 | 0.66 | 0.084 | 0.028 | 0.012 | 0.037 | 0.009 |
| X3 | 2.16 | 0.63 | 0.087 | 0.039 | 0.008 | 0.010 | 0.004 |
| X5 | 1.76 | 0.62 | 0.089 | 0.045 | 0.018 | 0.003 | 0.0009 |
| X6 | 1.44 | 0.63 | 0.092 | 0.031 | 0.015 | 0.003 | 0.0008 |
| X7 | 1.80 | 0.68 | 0.089 | 0.024 | 0.022 | 0.003 | 0.0009 |

[a] No enough sample for analysis

Table 4: The relative amounts (%) of nitrate, sulfate and ammonia in TSP or $PM_{2.5}$ in postfrontal air during dust storms at dust downwind Chinese cities.

| Site | $SO_4^{2-}$ | $NO_3^-$ | $NH_4^+$ | Remarks | References |
|------|------|------|------|---------|------------|
| Xi'an | 0.91 | 0.22 | NA | TSP (May 1, 2014) | This study |
| Xi'an | 1.05 | 0.21 | 0.09 | TSP (March 9, 2013) | Wang et al. (2014) |
| Beijing | 0.75 | 0.08 | 0.05 | TSP (March 22, 2002) | Zhao et al. (2007) |
| Shanghai | 2.7 | 1.4 | 0.7 | $PM_{2.5}$ (March 20-21, 2010) | Wang et al. (2013) |

**Figure caption**

Figure 1: Location of the sampling sites. Also shown is the Chinese Loess Plateau.

Figure 2: Surface pressure, temperature, relative humidity (RH), and wind during the sampling periods at the desert site (a: 2014/04/24 04:00-15:00 BST) and at Xi'an site (b: 2014/05/01 04:00-12:00 BST). Also shown are the front passage (shaded bars) and the durations of sample collections (Table 1).

Figure 3: Concentrations of mass, $SO_4^{2-}$, $NO_3^-$, $NH_4^+$ and $Ca^{2+}$ at Xi'an site during the dust passage. Data for samples between 11:27 and 12:27 BST are not included since the sampler was blown down by wind. The relative amounts (R. M.: the ratios of the ion concentrations to the total mass concentration in percentage) of these ions are also illustrated.

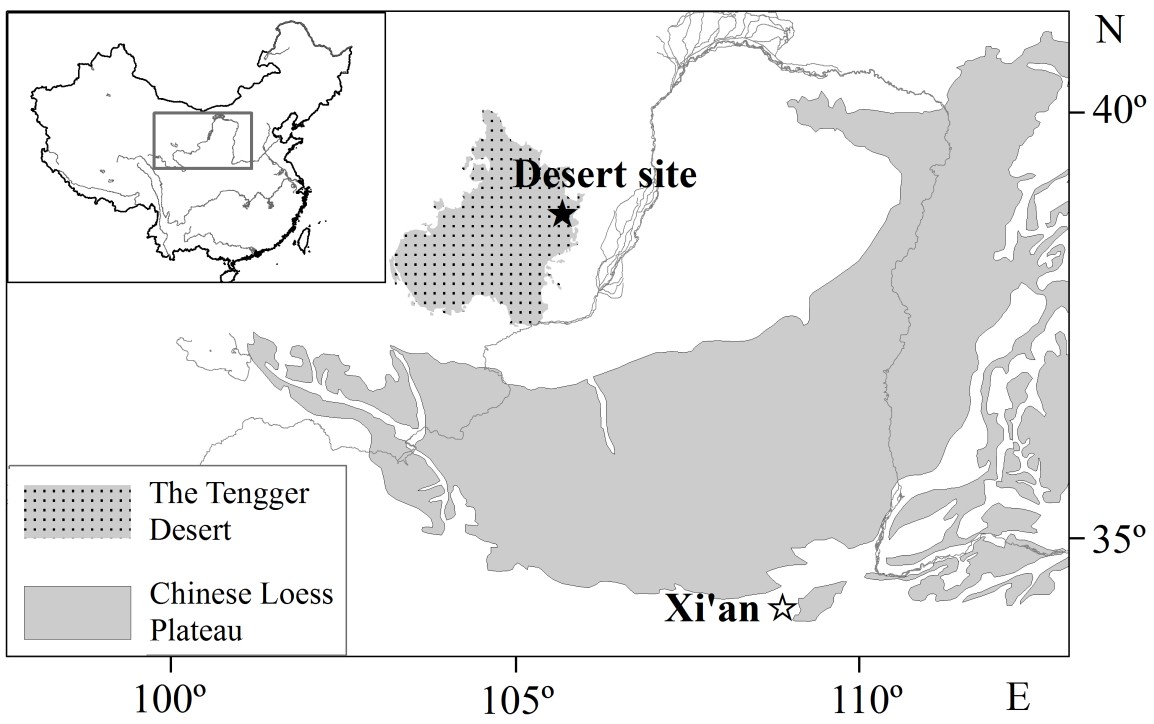

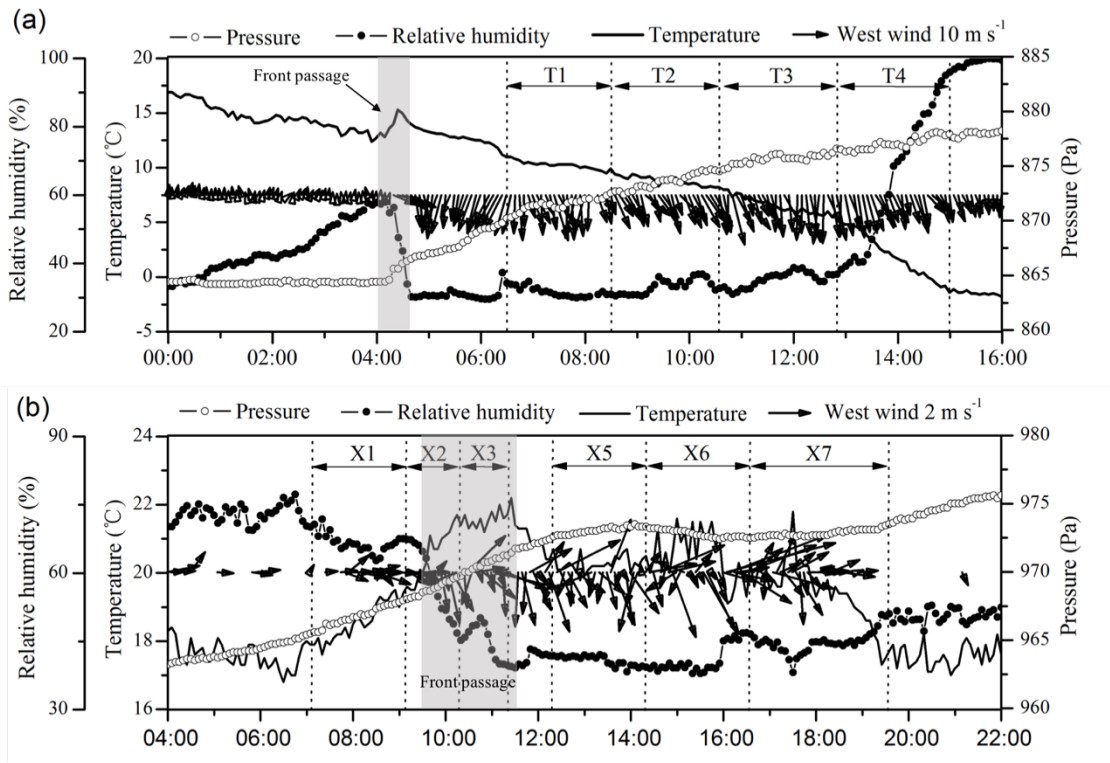

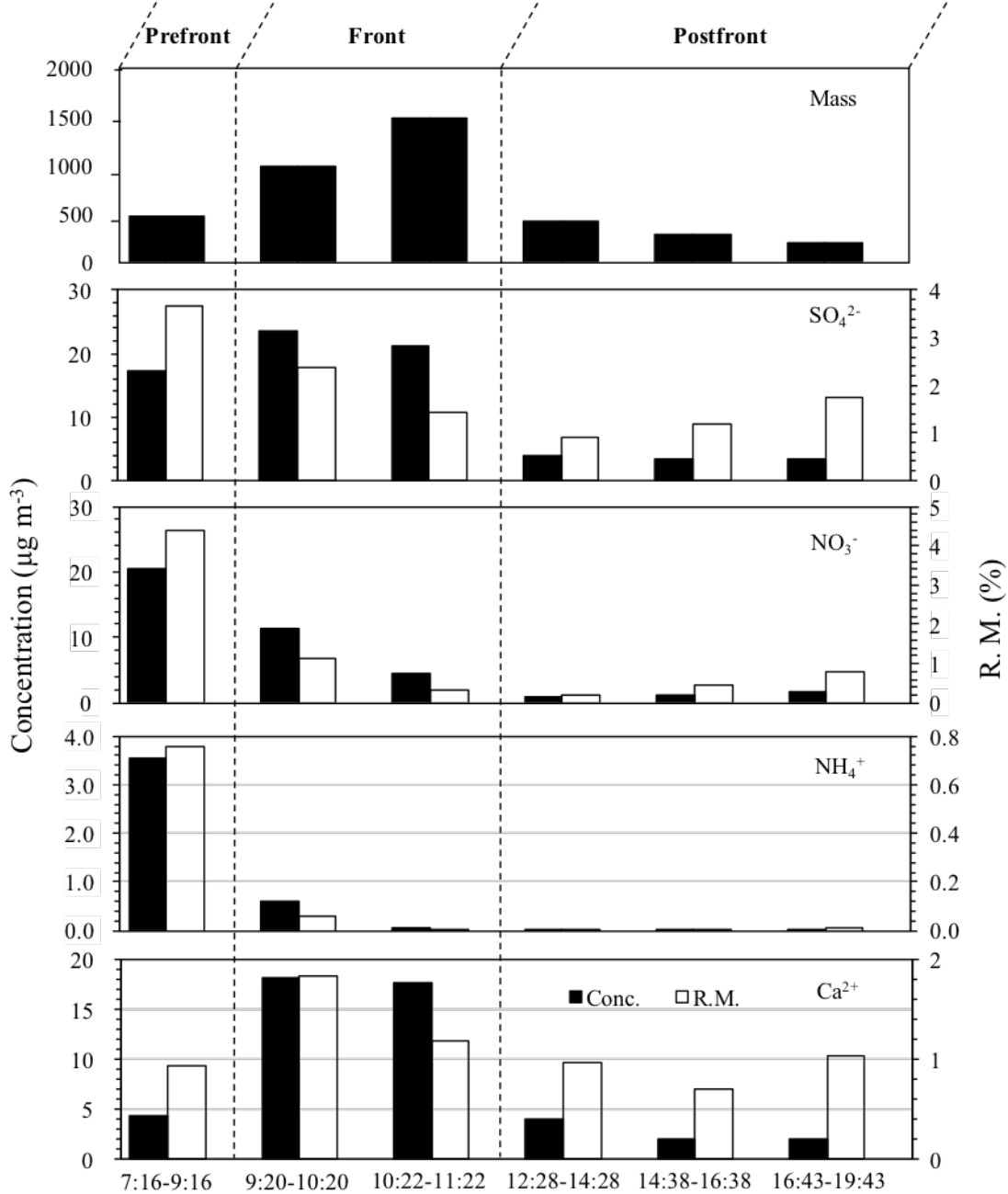