# Peer review of "Limited production of sulfate and nitrate on front-associated dust storm particles moving from desert to distant populated areas in northwestern China"

_Atmospheric Chemistry and Physics, 2016_

## Referee Comment (RC1) · B. Huebert (Referee) · 23 Dec 2016

The authors have made a handful of bulk aerosol composition measurements near-source and downwind (urban) of dust storms, separated by several days. (The transit time of airmasses between the sites would be about 6 hours.) They use concentration- and elemental ratio-differences between these locations to infer the formation or uptake of nitrate and sulfate with time.

They undertake a kind of Lagrangian analysis, as if the desert dust they first sampled had moved to Xian by the time they sampled there. (It had in fact moved past the downwind sampling site several days earlier.) The increase in the $NO_3$/Ca ratio, in particular, is used to infer a nitrate formation or uptake rate. To compute a trend,

one would need more samples and time-coordinated sampling. The uncertainty in the representativeness of the samples is at least as large as the apparent observed upwind/downwind differences. The high natural levels of sulfate in the dust (12%) no doubt vary, adding uncertainty to the inference of a trend.

There also aren't enough pieces of data to compute a defendable rate of ion formation on the dust. The experiment was poorly-posed to do so. Since only two sites were involved, it is impossible to infer nitrate increase over the desert vs nitrate picked up upon the dust's arrival in the urban area, based on their observations.

Line 22, Section 2: Here they argue that there should be no pollution in the samples, but in the Conclusions they suggest otherwise; indeed one cannot sample in an urban area and expect to avoid all pollution.

Typo: there is no April 31st.

Lines 12-13, P 5: Even though I have not seen the supplementary figures, I would in principal disagree that the changes in dust particles during transport would be the same for each event. That would need to be shown.

Lines 20-25, page 7: This is one of the fundamental problems with trying to interpret this data. They have no way to distinguish between sulfate from pollution and sulfate in the soil/dust itself.

Page 9, lines 19-21: "...very different from the conclusions of this study." What evidence is there that this study's "enhanced" (for purposes of discussion) nitrate was collected in transit vs from the populated area near the sampler? I believe this study's Conclusions are unsupported.

Page 9, line 30-31: Yes, prefrontal air is much more polluted than postfrontal air. But that doesn't prove that the postfrontal air is free of contamination. The postfrontal air is still moving across a landscape containing sources, especially near the sampling site in Xian. How rapidly would urban nitrate be formed, relative to the sampling interval in

the postfrontal air?

Furthermore, since there was only bulk sampling we don't know for sure that all the nitrate was even on the coarse (dust) mode. Their observations are simply too few and too limited in type to advance our understanding of the uptake of sulfate and nitrate by desert dust.

Page 10, line 4: Briefly explain "Peak 1" or don't mention it.

I really like most of the discussion on page 10, which addresses a way of identifying urban vs desert influences on dust composition using trace metals. Unfortunately this study only measured Ca, which is present in both desert and urban dust, so their conclusions can't benefit from this discussion.

For the reasons above, I recommend this paper be declined.

---

## Author Comment (AC1) · 12 Mar 2017

**huebert@hawaii.edu**

*The authors have made a handful of bulk aerosol composition measurements near source and downwind (urban) of dust storms, separated by several days. (The transit time of air masses between the sites would be about 6 hours.) They use concentration and elemental ratio-differences between these locations to infer the formation or uptake of nitrate and sulfate with time.*
*They undertake a kind of Lagrangian analysis, as if the desert dust they first sampled had moved to Xian by the time they sampled there. (It had in fact moved past the downwind sampling site several days earlier.) The increase in the $NO_3^-/Ca$ ratio, in particular, is used to infer a nitrate formation or uptake rate. To compute a trend, one would need more samples and time-coordinated sampling. The uncertainty in the representativeness of the samples is at least as large as the apparent observed upwind/downwind differences. The high natural levels of sulfate in the dust (12%) no doubt vary, adding uncertainty to the inference of a trend.*
**Response:** We want to make clear that (1) the natural level of sulfate we show in this paper was around 1.2% (Table 2) and was not 12%, and (2) we did not use the $NO_3^-/Ca$ ratio to infer a nitrate formation or uptake rate. Please check the contents of the manuscript again. For the formation of sulfate on dust storm particles, the result we observed was that the concentration levels of sulfate were similar at the two site (the level was 0.91% at the urban site; Table 3).

Even if we consider all the sulfate observed at the urban site was produced via reactions on particle surface, the production was still much smaller than that in polluted urban atmosphere. So we consider that sulfate was hardly produced. For the formation of nitrate, we found a small increase (the level was 0.12% at the desert site and 0.22% at the urban site), and then we used the recent numerical scheme of nitrate formation on dust particles (Fairlie et al. 2010) to estimate (with the conditions of possible nitric concentration, dust concentration, and the history of the air parcel) if the production of nitrate on dust particles during the dust travel was consistent with the level we observed at the urban site. We found the estimated one and the observed one were in the same order (Page 8 Line 18 – Page 9 Line 8). In fact, similar to sulfate, even though we consider all the nitrate observed at the urban site was produced via surface reactions, the production was still very smaller than that in polluted urban atmosphere.

Fairlie, T. D., Jacob, D. J., Dibb, J. E., Alexander, B., Avery, M. A., van Donkelaar, A. and Zhang, L.: Impact of mineral dust on nitrate, sulfate, and ozone in transpacific Asian pollution plumes, *Atmos. Chem. Phys.*, 10(8), 3999–4012, doi:10.5194/acp-10-3999-2010, 2010.

***There also aren't enough pieces of data to compute a defendable rate of ion formation on the dust. The experiment was poorly-posed to do so. Since only two sites were involved, it is impossible to infer nitrate increase over the desert vs nitrate picked up upon the dust's arrival in the urban area, based on their observations.***

**Response:** We do not think that data from more or less dust cases at the desert site and at the urban site are the key issues, although data from more cases are better. The key point is whether we get the common understandings from the data, no matter the data are more or less. There are a large number of published papers on the formation of sulfate and nitrate on dust particles observed in urban areas in mainland China. To the extent of what we can find, we have carefully checked all data in published literatures on the formation in dust storm particles in postfrontal air. We confirmed the common result as we describe in the manuscript: the production of the two salts on dust storm particles in postfrontal air was limited (some reference results in Table 3). So we think, even we increase the cases of dust observation at the urban site, we will encounter similar results.

For the data at the desert site, to the extent of our knowledge, the results reported in this study are the only data from a series samples that were carefully collected at a short time resolution from a dust storm at a desert dust dune all over the world. Yes, more case data will be better. Unfortunately, we failed in getting more high quality series of data from dust storms as we show in the manuscript, except for some pieces of data, due to technique problems.

For compensating this lack, we have carefully checked all published papers of studying nitrate and sulfate in dust from the Chinese Gobi Desert at observational sites in or close to the Tengger desert, and checked the data from samples that were considered to be dust storm particles with no anthropogenic pollution. We confirmed that the nitrate concentration from any dust storm samples in those papers was always very small and not very different from we encountered in this study (Table R1). So we consider our result on nitrate we observed at the desert can represent the common level of nitrate there. In the revision, we will add the range of nitrate concentration in desert dust plumes which were reported in published literatures to show

this point.

Table R1. The relative amounts (%) of nitrate and sulfate in dust samples from the Gobi Desert

| Study sites | Size fractions | $NO_3^-$ | $SO_4^{2-}$ | References |
|---|---|---|---|---|
| Ejin Qi, Badain Jaran desert [a] | TSP | 0.04 | 0.63 | Mori et at., 2002 |
| Sonid Youqi-Huade-Zhangbei | TSP | 0.025 | 0.46 | Mori et at., 2003 |
| Gobi desert [b] | $PM_{10}$ | 0.084 | 0.47 | Dong et al., 2016 |
| Tonggunao'er | TSP | 0.12±0.11 | 1.2±0.1 | This study |

[a] Estimated from regressions of aerosol chemical composition on distance from the kosa source. [b] Developed based on local measurement data collected by Huang et al. (2010).

Mori, I., Nishikawa, M., Quan, H., & Morita, M. (2002). Estimation of the concentration and chemical composition of kosa aerosols at their origin. *Atmos. Environ.*, *36*(29), 4569–4575, doi: 10.1016/S1352-2310(02)00489-2.

Mori, I., Nishikawa, M., Tanimura, T., & Quan, H. (2003). Change in size distribution and chemical composition of kosa (Asian dust) aerosol during long-range transport. *Atmos. Environ.*, *37*(30), 4253–4263, doi: 10.1016/S1352-2310(03)00535-1.

Dong, X., Fu, J. S., Huang, K., Tong, D., and Zhuang, G. (2016). Model development of dust emission and heterogeneous chemistry within the Community Multiscale Air Quality modeling system and its application over East Asia, Atmos. Chem. Phys., 16, 8157-8180, doi:10.5194/acp-16-8157-2016.

***Line 22, Section 2: Here they argue that there should be no pollution in the samples, but in the Conclusions, they suggest otherwise; indeed one cannot sample in an urban area and expect to avoid all pollution.***

**Response:** In this section, we concluded that there should be no pollution in the postfrontal samples of our study. Although the usage of "no" makes the meaning too absolute (we will decrease the tone in the revision), this does not contradict the conclusion that significant sulfate and nitrate in dust storm periods in China reported in previous studies were likely produced on locally-emitted and urban mineral particles. The reasons are that the separation of the prefrontal pollutants and the postfrontal dust plume was not considered and/or dust samples were not collected from postfrontal air only in those previous studies.

Yes, it is impossible to completely avoid pollution during any sample collection in an urban area. However, the question here is if the pollution is severe enough to lead to a considerable production of sulfate and nitrate on dust particles. The purpose of this study is to answer this question. As we mentioned in the manuscript, if the postfrontal samples were considerably polluted, there should have been some levels of ammonia (a common anthropogenic anion in urban air). The fact is that $NH_4^+$ concentration in the postfrontal air was lower than the detection limit in the first sample and increased slightly in the second and third samples. We also analyzed Zn and Pb, which are usually considered as anthropogenic trace elements in urban air. Their ratios to Fe in the dust in the postfrontal air were significantly lower than those in the prefrontal air and were very close to those in the desert air (Table R2), indicating that there should not be considerable pollutants in the samples.

Table R2 The ratios of Ca, Fe, Ti, Mn, Ba, Zn and Pb to Fe in aerosol samples at two sampling sites

| Samples | Ca/Fe | K/Fe | Ti/Fe | Mn/Fe | Ba/Fe | Zn/Fe | Pb/Fe |
|---|---|---|---|---|---|---|---|
| Tengger Desert (April 24, 2014) | | | | | | | |
| T1 | 1.47 | 0.54 | 0.084 | 0.023 | 0.013 | 0.003 | 0.0014 |
| T2 | 1.47 | 0.55 | 0.082 | 0.023 | 0.013 | 0.0023 | 0.0011 |
| T3 | 1.57 | 0.57 | 0.086 | 0.024 | 0.012 | 0.002 | 0.0009 |
| Xi'an (May 1, 2014) | | | | | | | |
| X1[a] | NA | NA | NA | NA | NA | NA | NA |
| X2 | 1.86 | 0.66 | 0.084 | 0.028 | 0.012 | 0.037 | 0.009 |
| X3 | 2.16 | 0.63 | 0.087 | 0.039 | 0.008 | 0.010 | 0.004 |
| X5 | 1.76 | 0.62 | 0.089 | 0.045 | 0.018 | 0.003 | 0.0009 |
| X6 | 1.44 | 0.63 | 0.092 | 0.031 | 0.015 | 0.003 | 0.0008 |
| X7 | 1.80 | 0.68 | 0.089 | 0.024 | 0.022 | 0.003 | 0.0009 |

[a] No enough sample for analysis

***Typo: there is no April 31st.***
**Response:** It is April 30th. We will correct it in the revised version.

***Lines 12-13, P5: Even though I have not seen the supplementary figures, I would in principal disagree that the changes in dust particles during transport would be the same for each event. That would need to be shown.***
**Response:** In the supplements, we show the back-trajectory routes from the desert site and the Xi'an site during two dust storm periods (Figure S2 and Figure S3) and also the vertical thermodynamic structure of postfrontal dust plumes (Figure S6) when the dust samples were collected. The figures show that the two dust storms were really very similar according to their transporting routes and thermodynamic structures. Since these data are from public sites and other simulations, we do not think that it is a good idea to show them in the main body of the manuscript.

Yes, it is not absolutely correct that "the changes in dust particles during transport would be the same for each event", and every dust storm must be more or less different from another dust storm. However, this does not mean we cannot find new understandings from a single dust storm which are common for dust storms. A single dust storm should have some common characteristics in a number of dust storms from the same desert. For your convenience to read them, we show the figures here to illustrate the similarities of the transport and the vertical thermodynamic structures of the two dust storms from which we collected samples. Please do us a favor to check the similarities between the two dust storms.

[Figure]

Figure S2: Backward trajectories from the desert site (2014/04/24) and Xi'an site (2014/05/01) from the HYSPLIT model (www.arl.noaa.gov/HYSPLIT.php). (BST = GMT + 08:00)

[Figure]

Figure S3: CFORS model output for boundary layer (surface - 1000m) dust concentration (μg/m$^3$, color in log scale) and wind vector at 1000m of East Asia during the sampling periods at desert site (a) and Xi'an (b). (http://www-cfors.nies.go.jp/~cfors/index-j.html) (JST = GMT + 09:00)

[Figure]

Figure S6: Vertical profiles of virtual potential temperature near the surface at Yinchuan (38.48°N, 106.21°E), the WMO sounding station closest to the desert site, and at Jinhe (34.43°N, 108.97°E), a suburb place of Xi'an, before and after dust occurrence at the two places. The profiles were from the homepage of Atmospheric Soundings of the University of Wyoming (http://weather.uwyo.edu/upperair/sounding.html). Dust occurred at the desert site on the morning of April 24, 2014, and the sample collection was held between 06:30 and 15:00 BST on April 24. Dust occurred at Xi'an site on the morning of May 1, 2014, and the sample collection was held between 07:00 and 19:00 BST on May 1.

***Lines 20-25, page 7: This is one of the fundamental problems with trying to interpret this data. They have no way to distinguish between sulfate from pollution and sulfate in the soil/dust itself.***

**Response:** Yes, it could be absolutely said that there is "no way to distinguish between sulfate from pollution and sulfate in the soil/dust itself". However, what we are discussing here is if

the small level of sulfate observed at the urban site (0.9%) was considerably larger than the level at the desert site (1.2%) and if the production of sulfate by surface reactions on dust particles during the particle travel was substantially large and has to be considered. Even the 0.9% of sulfate was totally from anthropogenic pollution, this does not contradict our conclusion that the production was limited. We do not have a reason to ignore the part of sulfate of mineral origin (1.2% in the present study) in the dust particles.

***Page 9, lines 19-21: "...very different from the conclusions of this study." What evidence is there that this study's "enhanced" (for purposes of discussion) nitrate was collected in transit vs from the populated area near the sampler? I believe this study's Conclusions are unsupported.***
**Response:** Below-detection-limit ammonia and unenriched Zn and Pb (we will add the data in the revision) relative to mineral dust in the postfrontal air indicate that the nitrate was impossibly explained by possible emissions from the populated area near the sampler. So we consider the nitrate was produced during the transport, although the amount was very limited in comparison with that in polluted urban atmosphere. Even though some of the "enhanced" nitrate was from the populated area near the sampler, the production of nitrate should be very small in comparison with that in polluted air, which supports our conclusion. Please also see our response to your next comment. In addition, it is very hard for us to believe the limited nitrate was produced in the last moment only before we collected the particles, because the conversion of background-like nitric acid to particle surface in dust air during the transport according to our estimation can, to a large extent, account for the nitrate production. Would you mind to give more details on why you disagree our conclusions, in order for us to understand accurately why you believe "the conclusions are unsupported".

***Page 9, line 30-31: Yes, prefrontal air is much more polluted than postfrontal air. But that doesn't prove that the postfrontal air is free of contamination. The postfrontal air is still moving across a landscape containing sources, especially near the sampling site in Xian. How rapidly would urban nitrate be formed, relative to the sampling interval in the postfrontal air?***
**Response:** We didn't attempt to show "that the postfrontal air is free of contamination" and we never say that in the manuscript. We show that the production of sulfate and nitrate on dust storm particles were limited. With our data, we estimated the rate of nitrate formation in the postfrontal dust when the particles travelled from the desert area to the urban area, as we show in the manuscript. The adiabatic state of the postfrontal dust plume was kept during the travel. So the rate was very small. Although there should be emissions of anthropogenic pollutants from local areas where the plume passed, the emitted amount was not large enough to influence the dust plume. Otherwise, gaseous pollutants such as $SO_2$ and $NO_2$ would not have decreased to very low levels. The major reason should be that the movement of the postfrontal air was relatively very fast, in comparison with prefrontal air.

***Furthermore, since there was only bulk sampling we don't know for sure that all the nitrate was even on the coarse (dust) mode. Their observations are simply too few and too limited in type to advance our understanding of the uptake of sulfate and nitrate by desert dust.***

**Response:** Currently, we do not have size-differentiated data for the formation of the salts to give a deeper discussion on in what size ranges of dust particles the salts were produced. However, as we mentioned in previous responses, even we consider all nitrate and sulfate were produced on the dust storm particles, the production of the salts was still very limited, which does not contradict our conclusions. Yes, it is true our methods are not advanced and the observations did not have many case data. Repeatedly, we think that the key point should be if our results support our conclusions, no matter whether the methods are advanced or not, and how many samples we have. To the extent of our knowledge, we did not find data that contradict our data and conclusions. The differences between our data and published data were reasonably explained in the discussion of the manuscript.

*Page 10, line 4: Briefly explain "Peak 1" or don't mention it.*
**Response:** Peak 1 in the study of Zhao et al. (2007) referred to the period of the highest loading of mass during the dust period.

In the revision, we will remove "Peak 1" from the text. The sentence "…, the mineral/TSP ratios in samples with the highest TSP loading (Peak I described in that study) were significantly lower than those in samples collected after the occurrence of maximum aerosol loading, indicating that the samples around Peak I were not dust particles from desert areas only." will be revised in the revision

*I really like most of the discussion on page 10, which addresses a way of identifying urban vs desert influences on dust composition using trace metals. Unfortunately, this study only measured Ca, which is present in both desert and urban dust, so their conclusions can't benefit from this discussion.*
**Response:** Thank you very much for your encouragement. In this section, we discussed the indictors of discriminating desert dust from urban aerosols in some previous studies to explain why some studies encountered the result that some "dust samples" contained substantial sulfate and nitrate. We emphasize that $Ca^{2+}$ is present in both desert dust and urban mineral particles and is not as a good indicator for discriminating desert dust from urban aerosols. Dust samples in studies using $Ca^{2+}$ as the indicator of the presence of desert dust could be a mixture of long-distance transported dust particles and locally- and regionally-originated aerosols. The mixture caused the conclusion in some previously published papers that dust particles significantly enhanced the formation of sulfate and nitrate when dust plumes advected over urban areas. So the discussion helps to elucidate the discrepancy between the results of different studies as mentioned in the Introduction.

In our study, we did not use metals as indicators to discriminating desert dust from urban aerosol. We divided the sampling periods into three stages: prefrontal, frontal and postfrontal air. We found that the production of nitrate and sulfate in samples dominated by desert dust particles (in the postfrontal air) was very limited and we explained the results based on the adiabatic movement of the postfrontal dust plume.

In addition, we also measured other metals. In the revised manuscript, we will add the results of two common anthropogenic trace elements Zn and Pb, as mentioned in previous responses. The ratios of them to Fe in postfrontal dust particles were very close to those in the desert air, and much smaller than those in the prefrontal air (Table R2), further suggesting the

limited influence of pollution on desert dust particles.

Thank you very much again for our careful review of our submission. Your any further comments are welcome.

---

## Referee Comment (RC2) · Anonymous Referee #2 · 16 May 2017

General comments:

A number of laboratory and field studies have proved that Asian dust particles readily promoted the formation of sulfate and nitrate when the lofted dust plumes transported across urban areas under high RH and elevated levels of reactive trace gases (i.e. $SO_2$, $NO_x$, $O_3$ and OH radicals). This would significantly alter the physical and chemical properties of dust aerosol and subsequent climate change on regional scale.

The authors carried out a series of particle samplings at the Tengger desert (06:30–15:00 BST on April 24, 2014) and downwind Xi'an city (07:00–19:00 BST on May 1, 2014) during two independent dust storms. Combination of HYSPLIT backward trajectories model and CFORS simulation, they showed that the two dust events originated

from the same source regions and had similar transport routes. They compared the concentrations and mass fraction of chemical components (i.e. sulfate, nitrate, ammonia, and elemental ratios) in dust particles at two sites during the prefrontal, frontal, and postfrontal air parcels, and indicated that the production of sulfate and nitrate on front-associated dust particles was limited when the dust moved from desert sources to populated areas in northwest China. The result of this manuscript seems to be reasonable in spite of limited in-situ sampling data, which is completely different from the other previous studies. Different scientific viewpoints should be encouraged to promote the understanding of interplay between mineral dust and atmospheric chemistry. Therefore, I recommend this manuscript is accepted and published in the journal of ACP after some revisions.

Specific comments:

1. Abstract, Page 2, lines 12–14: "The significant sulfate and nitrate reported in dust-associated samples in previous studies were more likely produced on locally-emitted and urban mineral particles or from soil-derived sulfate rather than being produced via chemical conversions on desert dust particles."

Conclusion, Page 11, lines 2–4: "Significant sulfate and nitrate in dust storm periods in China reported in previous studies were likely produced on locally-emitted and urban mineral particles, in addition to soil-derived sulfate, and they were unlikely produced via chemical conversions on dust particles from deserts."

Comment: I think there is no enough evidence for the manuscript to demonstrate this conclusion. Because the dry condition (RH<40

2. Page 4, lines 21–22: "This sample collection ensured that mineral particles collected on the filters were dust particles from the desert and there should be no influence of anthropogenic pollutants from the village or the city considered in the samples."

Comment: The evidence provided by this manuscript could not fully support this sentence. Please reconsider again.

3. Page 6, lines 15–17: "The cold fronts are the boundaries between the local or regional anthropogenic-polluted air and the long-distance transported air because the movement of air on a synoptic scale is approximately adiabatic, i.e. the air is hardly mixed with thermodynamically-different air it meets."

Comment: I don't agree with this viewpoint about "the air is hardly mixed with thermodynamically-different air it meets". In terms of meteorology, the warm and humid air mass is readily lifted and the weather process (e.g., strong wind and cooling weather, rainfall or snow) often changes dramatically on the border of frontal system when a cold front passes over. As shown in Figure 2a, the RH increased sharply from 40

4. Page 7, lines 26–33: "At the desert site, NO3- concentration in dust samples was 4-6 ugm-3 and the average was 5 ugm-3. The relative amount of NO3- range between 0.11

Comment: At the Tengger desert site, NO3- concentration in dust samples was 4-6 ugm-3 (with the average value and fraction of 5 ugm-3 and 0.12

5. Page 18, Table 2; Page 19, Table 3: The authors sampled the concentrations of TSP (total suspended particulates) and analyzed the chemical components (i.e. sulfate, nitrate, and ammonia) in TSP at the Tengger desert and Xi'an sites.

Comment: Please explain why did you sample the concentrations of TSP, instead of PM10 or PM2.5. It is well known that most of the coarse-size dust particles (radii > 10 um) generally settle near the source region on account of large gravitational deposition velocity, whereas the finer dust particles (radii < 10 um) are transported more efficiently to the downstream areas. And the concentrations of TSP (meaning coarse-size particle with radii>10 um) in Xi'an city should include the local source emissions (e.g., engines of vehicles, road dust, and construction dust; Page 4, lines 27-29) that increases the

TSP concentrations, but may decrease the relative mass fraction of sulfate, nitrate, and ammonia. Inferred from Page 6, Lines 1-7, the mass concentrations of TSP and sulfate are about 425 ugm-3 and 17 ugm-3 at Xi'an before the dust arrival (i.e. prefrontal air). In the postfrontal air, the mass concentrations of sulfate are 3.8, 3.5, and 3.4 ugm-3, respectively, right after, two hour after, and four hours after the passage of cold front (means slight variations), whereas the corresponding TSP concentrations are 422, 318, and 189 ugm-3 (shows large variations). Clearly, relative mass fractions of sulfate reduce.

Minor comments:

1. Abstract, Page 2, lines 3–4: "but the production was very inefficient in other studies."

Comment: Please give the quoted literature.

2. Page 3, line 8: "RH"

Comment: Change to "relative humidity (RH)"

3. Page 10, line 4: "mineral/TSP ratios"

Comment: Change to "mineral/TSP (total suspended particulates) ratios"

Please also note the supplement to this comment:
http://www.atmos-chem-phys-discuss.net/acp-2016-853/acp-2016-853-RC2-supplement.pdf
* * *
[Figure]

**Supplement:**

**General comments:**

A number of laboratory and field studies have proved that Asian dust particles readily promoted the formation of sulfate and nitrate when the lofted dust plumes transported across urban areas under high RH and elevated levels of reactive trace gases (i.e. $SO_2$, NOx, $O_3$ and OH radicals). This would significantly alter the physical and chemical properties of dust aerosol and subsequent climate change on regional scale.

The authors carried out a series of particle samplings at the Tengger desert (06:30–15:00 BST on April 24, 2014) and downwind Xi'an city (07:00–19:00 BST on May 1, 2014) during two independent dust storms. Combination of HYSPLIT backward trajectories model and CFORS simulation, they showed that the two dust events originated from the same source regions and had similar transport routes. They compared the concentrations and mass fraction of chemical components (i.e. sulfate, nitrate, ammonia, and elemental ratios) in dust particles at two sites during the prefrontal, frontal, and postfrontal air parcels, and indicated that the production of sulfate and nitrate on front-associated dust particles was limited when the dust moved from desert sources to populated areas in northwest China. The result of this manuscript seems to be reasonable in spite of limited in-situ sampling data, which is completely different from the other previous studies. Different scientific viewpoints should be encouraged to promote the understanding of interplay between mineral dust and atmospheric chemistry. Therefore, I recommend this manuscript is accepted and published in the journal of ACP after some revisions.

**Specific comments:**

1. **Abstract**, Page 2, lines 12–14: "The significant sulfate and nitrate reported in

dust-associated samples in previous studies were more likely produced on locally-emitted and urban mineral particles or from soil-derived sulfate rather than being produced via chemical conversions on desert dust particles."

**Conclusion**, Page 11, lines 2–4: "Significant sulfate and nitrate in dust storm periods in China reported in previous studies were likely produced on locally-emitted and urban mineral particles, in addition to soil-derived sulfate, and they were unlikely produced via chemical conversions on dust particles from deserts."

**Comment:** I think there is no enough evidence for the manuscript to demonstrate this conclusion. Because the dry condition (RH<40%) and low mass concentrations of trace gases (i.e. $SO_2$ and $NO_2$) were observed in Xi'an during the dust-storm episode, which didn't favor the formation of sulfate and nitrate on the surface of mineral particles. However, these are only a few cases. The authors didn't show the results when dust storm transported across the other polluted areas with high RH conditions and high levels of trace gases.

2. Page 4, lines 21–22: "This sample collection ensured that mineral particles collected on the filters were dust particles from the desert and there should be no influence of anthropogenic pollutants from the village or the city considered in the samples."

**Comment:** The evidence provided by this manuscript could not fully support this sentence. Please reconsider again.

3. Page 6, lines 15–17: "The cold fronts are the boundaries between the local or regional anthropogenic-polluted air and the long-distance transported air because the movement of air on a synoptic scale is approximately adiabatic, i.e. the air is hardly mixed with thermodynamically-different air it meets."

**Comment:** I don't agree with this viewpoint about "the air is hardly mixed with thermodynamically-different air it meets". In terms of meteorology, the warm and humid air mass is readily lifted and the weather process (e.g., strong wind and cooling weather, rainfall or snow) often changes dramatically on the border of frontal system

when a cold front passes over. As shown in Figure 2a, the RH increased sharply from 40% at 13:00 BST to 100% at 16:00 BST, which indicated clearly that a rainfall or snow process took place at Tengger desert (also see Page 4, line 15 and Table 1). The cold fronts are dominated and accompanied strong winds intensify the diffusions of local air pollutants.

4. Page 7, lines 26–33: "At the desert site, $NO_3^-$ concentration in dust samples was 4-6 $\mu gm^{-3}$ and the average was 5 $\mu gm^{-3}$. The relative amount of $NO_3^-$ range between 0.11% and 0.12%, and the average was 0.12%. …Right after the passage of the cold front (the first sample in the postfrontal air), the concentration of $NO_3^-$ was 0.9 $\mu gm^{-3}$ and it occupied 0.2% of the aerosol mass. The relative amount in this sample was about twice of that in the desert samples although it was the lowest in the samples at Xi'an site, indicating that nitrate had been produced on dust particles during their travel to Xi'an."

**Comment:** At the Tengger desert site, $NO_3^-$ concentration in dust samples was 4-6 $\mu gm^{-3}$ (with the average value and fraction of 5 $\mu gm^{-3}$ and 0.12%), which were much larger than that at Xi'an after the cold front (~0.9 $\mu gm^{-3}$, with the mass fraction ~0.2%). The higher mass fraction of $NO_3^-$ at Xi'an was ascribed to the low concentration of TSP (total suspended particulates, ~420 $\mu gm^{-3}$), and the TSP concentration in Tengger desert site was about 5000 $\mu gm^{-3}$. Although the relative amount of $NO_3^-$ at Xi'an was about twice of that in the desert samples, it couldn't indicate that nitrate had been produced on dust particles during their travel to Xi'an. Please explain this.

5. Page 18, Table 2; Page 19, Table 3: The authors sampled the concentrations of TSP (total suspended particulates) and analyzed the chemical components (i.e. sulfate, nitrate, and ammonia) in TSP at the Tengger desert and Xi'an sites.

**Comment:** Please explain why did you sample the concentrations of TSP, instead of $PM_{10}$ or $PM_{2.5}$. It is well known that most of the coarse-size dust particles (radii > 10 $\mu m$) generally settle near the source region on account of large gravitational

deposition velocity, whereas the finer dust particles (radii < 10 μm) are transported more efficiently to the downstream areas. And the concentrations of TSP (meaning coarse-size particle with radii>10 μm) in Xi'an city should include the local source emissions (e.g., engines of vehicles, road dust, and construction dust; Page 4, lines 27-29) that increases the TSP concentrations, but may decrease the relative mass fraction of sulfate, nitrate, and ammonia. Inferred from Page 6, Lines 1-7, the mass concentrations of TSP and sulfate are about 425 μgm$^{-3}$ and 17 μgm$^{-3}$ at Xi'an before the dust arrival (i.e. prefrontal air). In the postfrontal air, the mass concentrations of sulfate are 3.8, 3.5, and 3.4 μgm$^{-3}$, respectively, right after, two hour after, and four hours after the passage of cold front (means slight variations), whereas the corresponding TSP concentrations are 422, 318, and 189 μgm$^{-3}$ (shows large variations). Clearly, relative mass fractions of sulfate reduce.

**Minor comments:**

1. **Abstract**, Page 2, lines 3–4: "but the production was very inefficient in other studies."

**Comment:** Please give the quoted literature.

2. Page 3, line 8: "RH"

**Comment:** Change to "relative humidity (RH)"

3. Page 10, line 4: "mineral/TSP ratios"

**Comment:** Change to "mineral/TSP (total suspended particulates) ratios"

---

## Author Response (AR1)

June 8, 2017

Dear editor

Here, we submit the revised manuscript (acp-2016-853) entitled, "Limited production of sulfate and nitrate on front-associated dust storm particles moving from desert to distant populated areas in northwestern China" to the special issue of "Anthropogenic dust and its climate impact" of your journal *Atmospheric Chemistry and Physics*. We would like to thank the two referees for their careful and constructive reviews. Based the comments from the referees, we have revised carefully the manuscript.

Attached are the list of all relevant changes made in the manuscript, our point-by-point responses to all review remarks and a marked-up manuscript version in which all changes were marked with underlines.

Thanks for your help.

Yours sincerely,

Daizhou Zhang
* * *
Faculty of Environmental and Symbiotic Sciences
Prefectural University of Kumamoto
Kumamoto 862-8502, Japan
Tel:+81-(0)96-321-6712     Fax:+81-(0)96-384-6765
Email: dzzhang@pu-kumamoto.ac.jp
* * *
**List of all relevant changes made in the manuscript**

1. **In line 12-15 of Page 2:** The sentence **"The significant sulfate and nitrate reported in dust-associated samples in previous studies were more likely produced on locally-emitted and urban mineral particles or from soil-derived sulfate rather than being produced via chemical conversions on desert dust particles."** is changed to **"The significant sulfate and nitrate reported in storm-associated samples in previous studies were more likely from locally-emitted and urban mineral particles or from soil-derived sulfate, because the weather conditions in those studies indicated that the air from which the samples were collected very likely contained a lot of particles from local emissions."**

2. **In line 8 of Page 3:** "RH" was changed to "**relative humidity (RH)**"

3. In line 23-24 of Page 4: The last sentence of the second paragraph of subsection 2 "**This sample collection ensured that mineral particles collected on the filters were dust particles from the desert and there should be no influence of anthropogenic pollutants from the village or the city considered in the samples.**" was modified into **"This sample collection ensured that mineral particles collected on the filters were dominated by dust particles from the desert and possible influence of anthropogenic pollutants from the village or the city was suppressed."**

4. **In line 4 of Page 5:** "April 31" was corrected into "**April 30**".

5. **From line 31 of Page 5 to line 2 of Page 6**: The sentences **"An energy dispersive X-Ray fluorescence (ED-XRF) spectrometry (Epsilon 5 ED-XRF, PANalytical B. V., the Netherlands) was used to quantify elements in the samples of the remained parts of sample filters. Five crustal elements (K, Ca, Ti, Mn, Fe and Ba) and two common anthropogenic trace elements (Zn and Pb), were quantitatively determined. Analytical uncertainties, as checked by parallel analysis of the NIST standard reference material (SRM-2683), were about or less than 10% for the detected elements."** was added to describe how the elements were analyzed.

6. **In line 7-8 of Page 6:** We added "**These values were close to the levels of the relative mass ratios of sulfate in TSP or PM$_{10}$ in samples collected under dust conditions at the Gobi Desert which were reported in previous studies (Table S1).**"

7. **In line 25-26 of Page 6**: The sentence **"although some small-scale mixing might occur in the front"** is added after the sentence **"The cold fronts are the boundaries between the local or regional anthropogenically-polluted air and the long-distance transported air because the movement of air on a synoptic scale is approximately adiabatic, i.e.**

the air is hardly mixed with thermodynamically-different air it meets.**".

8. **In line 4-7 of Page 7:** We added the sentences "**Zn and Pb are two common anthropogenic trace elements in urban air. Their ratios to Fe in the dust samples in the postfrontal air were much lower than those in the prefrontal air and very close to those in the desert air (Table 3), further suggesting the limited influence of pollution on desert dust particles in the postfrontal air.**"

9. **In line 2 of Page 8:** "completely" was changed to **"mainly"**

10. **In line 10-12 of Page 8:** We added "**These values were close to the levels of the relative mass ratios of nitrate in TSP or PM$_{10}$ in samples collected under dust conditions at the Gobi Desert which were reported in previous studies (Table S1).**"

11. **In line 14 of Page 7**: **"had been"** was revised into **"was likely"**

12. **In line 22-25 of Page 8:** The descriptions **"Note this rate should be the maximum rate because not all the nitrate must have been produced on dust particles and the increase of the relative amount of nitrate during the movement of a dust plume from the desert to Xi'an could have been the consequence of possible different removal rates of dust particles and nitrate-containing particles.**" are added right after the descriptions of the estimated rate values of nitrate production on dust particles.

13. **In line 27 of Page 9:** "Table 3" was changed into "Table 4"

14. **In line 22 of Page 10:** "mineral/TSP" was changed to "**mineral/TSP (total suspended particulates) ratios**"

15. **In line 21-24 of Page 10:** "Peak 1" was removed from the text. **The sentence "…, the mineral/TSP ratios in samples with the highest TSP loading (Peak I described in that study) were significantly lower than those in samples collected after the occurrence of maximum aerosol loading, indicating that the samples around Peak I were not dust particles from desert areas only."** was changed to **"…, the mineral/TSP ratios in samples of the highest TSP loading were significantly lower than those in samples collected after the occurrence of maximum aerosol loading, indicating that the samples at the highest TSP moment were not dust particles from desert areas only."**

16. **In line 21-26 of Page 11**: **"Significant sulfate and nitrate in dust storm periods in China reported in previous studies were likely produced on locally-emitted and urban mineral particles, in addition to soil-derived sulfate, and they were unlikely produced via chemical conversions on dust particles from deserts."** is changed into (as an individual paragraph) **"Significant sulfate and nitrate in dust storm periods in China**

reported in previous studies were likely from locally-emitted and urban mineral particles, in addition to soil-derived sulfate, and they were unlikely produced via chemical conversions on dust particles from deserts. The major reason is that, in those studies, the air from which the samples were collected had been significantly influenced by local emissions. Without a proper evaluation of the contribution of sulfate and nitrate in the samples by locally-emitted and urban mineral particles, i.e., non-desert mineral particles, it is not safe to attribute the all detected sulfate and nitrate to the production on dust-storm particles.”

17. **In line 26-28 of Page 12:** the bibliography of Dong et al. (2016) was included in the reference list.

18. **In line 1 of Page 17:** The title of Table 1 was changed to **“Summary of weather conditions at the sample collection”**

19. **In Page 19: The table for mass ratios of Ca, Fe, Ti, Mn, Ba, Zn and Pb to Fe in the samples was added in the manuscript as Table 3, and the Table 3 in the last submission was changed into Table 4.**

20. **In the supplementary information: Table S1 was added.**

**Point-by-point response to all review remarks**

*Reviewer# Dr. Barry Huebert*
**The authors greatly appreciate the careful and detailed review by *Dr. Barry Huebert***
* * *
**Introduction:**
**The responses in letter are similar to our responses uploaded on 12 March 2017 to response the comments and questions by Dr. B. Huebert. In this letter, we added the descriptions of the revised parts (the parts in Bold and Underlined) in the manuscript upon the comments.**
* * *
*The authors have made a handful of bulk aerosol composition measurements near source and downwind (urban) of dust storms, separated by several days. (The transit time of air masses between the sites would be about 6 hours.) They use concentration and elemental ratio-differences between these locations to infer the formation or uptake of nitrate and sulfate with time.*
*They undertake a kind of Lagrangian analysis, as if the desert dust they first sampled had moved to Xian by the time they sampled there. (It had in fact moved past the downwind sampling site several days earlier.) The increase in the $NO_3^-/Ca$ ratio, in particular, is used to infer a nitrate formation or uptake rate. To compute a trend, one would need more samples and time-coordinated sampling. The uncertainty in the representativeness of the samples is at least as large as the apparent observed upwind/downwind differences. The high natural levels of sulfate in the dust (12%) no doubt vary, adding uncertainty to the inference of a trend.*

**Response:** We want to make clear that (1) the natural level of sulfate we showed in this paper was around 1.2% (Table 2) and was not 12%, and (2) we did not use the $NO_3^-$/Ca ratio to infer a nitrate formation or uptake rate. Please check the contents of the manuscript again. For the formation of sulfate on dust storm particles, the fact we encountered is that the concentration level of sulfate did not increased (the level was 0.91% at the urban site; Table 3). Even if we consider all the sulfate observed at the urban site were produced via surface reactions, the production was still much smaller than that in polluted urban atmosphere. So we considered that sulfate was hardly produced. For the formation of nitrate, we found a small increase (the level was 0.12% at the desert site and 0.22% at the urban site), and then we used the state-of-the-art numerical scheme of nitrate formation on dust particles (Fairlie et al. 2010) to estimate (with the conditions of possible nitric concentration, dust concentration, and the history of the air parcel) if the production of nitrate on dust particles was consistent with the level we observed at the urban site. We found they were in the same order (Page 8 Line 18 – Page 9 Line 8). In fact, similar to sulfate, even if we consider all the nitrate observed at the urban site were produced via surface reactions, the production was still very small than that in polluted urban atmosphere.

*There also aren't enough pieces of data to compute a defendable rate of ion formation on the dust. The experiment was poorly-posed to do so. Since only two sites were involved, it is*

*impossible to infer nitrate increase over the desert vs nitrate picked up upon the dust's arrival in the urban area, based on their observations.*

**Response:** We do not think that data from more or less dust cases at the desert site and at the urban site are the key issue, although data from more cases are better. The key point is if we get the common understandings from the data, no matter the data are more or less. There are a large number of published papers on the formation of sulfate and nitrate on dust particles observed in urban areas in mainland China. To the extent of what we can find, we checked all data in the published literatures on the formation on dust storm particles in post-frontal air. We confirmed the common results as we described in the manuscript: the production of the two salts on dust storm particles in postfrontal air was limited (some results in Table 3). So we think, even we increase the cases of dust observation at the urban site, we will encounter the same results. For the data at the desert site, to the extent of our knowledge, the results reported in this study are the only data from a series samples that were carefully collected at a short time resolution from a dust storm at a desert dust dune all over the world. Yes, more case data will be better. The fact is that we have been there several times and we failed in getting more series of data from dust storms except for some pieces of data due to technique problems, although this is not a scientific reason (*please imagine how can collect a series of dust samples at 2-hour time resolution at a dust dune in a desert area from a severe dust storm without electricity supply*). For compensating this problem, we carefully investigated all published papers of studying nitrate and sulfate in dust from the Gobi Desert at the observation sites in or near this desert area which used samples from routine observation sites near Chinese desert areas, and checked the data from samples that were really obtained under dust conditions or after dust storm arrival (some discussions in section 3.3) of desert dust. We confirmed that the concentration of sulfate and nitrate from the samples under dust conditions in those papers was close to or in the same order as our results in this study (Supplementary Table S1). So we consider our results on sulfate and nitrate at the desert can represent the common level of sulfate and nitrate of natural dust particles at the desert site.

**In the revision, we added that "These values were close to the levels of the relative mass ratios of sulfate in TSP or $PM_{10}$ in samples collected under dust conditions at the Gobi Desert which were reported in previous studies (Table S1)." at the end of 1st paragraph of subsection 3.1, where the concentration of sulfate in dust samples at the desert site was described.**

**For nitrate at the desert site, we added "These values were close to the levels of the relative mass ratios of nitrate in TSP or $PM_{10}$ in samples collected under dust conditions at the Gobi Desert which were reported in previous studies (Table S1)." at the end of 1st paragraph of subsection 3.2, where the concentration of nitrate in dust samples at the desert site was described.**

**In addition, the table mentioned in these descriptions was added in the supplementary information as Table S1. The bibliography of Dong et al. (2016) was also included in the reference list.**

Table S1. The relative mass ratio (%) of nitrate and sulfate in samples collected under dust conditions at the Gobi Desert reported in previous studies.

| Study sites | Size fractions | $NO_3^-$ | $SO_4^{2-}$ | References |
|---|---|---|---|---|

| | | | | |
|---|---|---|---|---|
| Ejin Qi, Badain Jaran desert [a] | TSP | 0.04 | 0.63 | Mori et at., 2002 |
| Sonid Youqi-Huade-Zhangbei | TSP | 0.025 | 0.46 | Mori et at., 2003 |
| Gobi desert [b] | $PM_{10}$ | 0.084 | 0.47 | Dong et al., 2016 |
| Tonggunao'er | TSP | 0.12±0.11 | 1.2±0.1 | This study |

[a] Estimated from regressions of aerosol chemical composition on distance from the dust source. [b] Based on the local measurement data reported by Huang et al. (2010).

Mori, I., Nishikawa, M., Quan, H., & Morita, M. (2002). Estimation of the concentration and chemical composition of kosa aerosols at their origin. *Atmos. Environ.*, *36*(29), 4569–4575, doi: 10.1016/S1352-2310(02)00489-2.

Mori, I., Nishikawa, M., Tanimura, T., & Quan, H. (2003). Change in size distribution and chemical composition of kosa (Asian dust) aerosol during long-range transport. *Atmos. Environ.*, *37*(30), 4253–4263, doi: 10.1016/S1352-2310(03)00535-1.

Dong, X., Fu, J. S., Huang, K., Tong, D., and Zhuang, G. (2016). Model development of dust emission and heterogeneous chemistry within the Community Multiscale Air Quality modeling system and its application over East Asia, Atmos. Chem. Phys., 16, 8157-8180, doi:10.5194/acp-16-8157-2016.

***Line 22, Section 2: Here they argue that there should be no pollution in the samples, but in the Conclusions, they suggest otherwise; indeed one cannot sample in an urban area and expect to avoid all pollution.***

**Response:** In this section, we concluded that there should be no pollution in the postfrontal samples of our study. Although the usage of "no" makes the meaning very absolute, this does not contradict the conclusion that significant sulfate and nitrate in dust storm periods in China reported in previous studies were likely produced on locally-emitted and urban mineral particles. The reasons are that the separation of the prefrontal pollutants and the postfrontal dust plume was not considered and/or dust samples were not collected from postfrontal air only in those previous studies.

**In the revision,** **this sentence was modified into "This sample collection ensured that mineral particles collected on the filters were dominated by dust particles from the desert and possible influence of anthropogenic pollutants from the village or the city was suppressed."**

Yes, it is impossible to completely avoid pollution during any sample collection in an urban area. However, the question here is if the pollution is severe enough to lead to a considerable production of sulfate and nitrate on dust particles. The purpose of this study is to answer this question. As we mentioned in the manuscript, if the postfrontal samples were considerably polluted, there should have been some levels of ammonia (a common anthropogenic anion in urban air). The fact is that $NH_4^+$ concentration in the postfrontal air was lower than the detection limit in the first sample and increased slightly in the second and third samples. We also analyzed Zn and Pb, which are usually considered as anthropogenic trace elements in urban air. Their ratios to Fe in the dust in the postfrontal air were significantly lower than those in the prefrontal air and were very close to those in the desert air (Table 3). So there should be unlikely considerable pollution in the samples.

Table 3 Mass ratios of Ca, Fe, Ti, Mn, Ba, Zn and Pb to Fe in the samples at the two sites

| Samples | Ca/Fe | K/Fe | Ti/Fe | Mn/Fe | Ba/Fe | Zn/Fe | Pb/Fe |
|---------|-------|------|-------|-------|-------|-------|-------|
| Tengger Desert (April 24, 2014) | | | | | | | |
| T1 | 1.47 | 0.54 | 0.084 | 0.023 | 0.013 | 0.003 | 0.0014 |
| T2 | 1.47 | 0.55 | 0.082 | 0.023 | 0.013 | 0.0023 | 0.0011 |
| T3 | 1.57 | 0.57 | 0.086 | 0.024 | 0.012 | 0.002 | 0.0009 |
| Xi'an (May 1, 2014) | | | | | | | |
| X1[a] | NA | NA | NA | NA | NA | NA | NA |
| X2 | 1.86 | 0.66 | 0.084 | 0.028 | 0.012 | 0.037 | 0.009 |
| X3 | 2.16 | 0.63 | 0.087 | 0.039 | 0.008 | 0.010 | 0.004 |
| X5 | 1.76 | 0.62 | 0.089 | 0.045 | 0.018 | 0.003 | 0.0009 |
| X6 | 1.44 | 0.63 | 0.092 | 0.031 | 0.015 | 0.003 | 0.0008 |
| X7 | 1.80 | 0.68 | 0.089 | 0.024 | 0.022 | 0.003 | 0.0009 |

[a] No enough sample for analysis

**In the revision, the table mentioned in these descriptions was added in the manuscript as Table 3, and Table 3 in the last submission was changed into Table 4.**
**We added the sentences "An energy dispersive X-Ray fluorescence (ED-XRF) spectrometry (Epsilon 5 ED-XRF, PANalytical B. V., the Netherlands) was used to quantify elements in the samples of the remained parts of sample filters. Five crustal elements (K, Ca, Ti, Mn, Fe and Ba) and two common anthropogenic trace elements (Zn and Pb), were quantitatively determined. Analytical uncertainties, as checked by parallel analysis of the NIST standard reference material (SRM-2683), were about or less than 10% for the detected elements." at the end of the last paragraph of subsection 2, to describe how the elements were analyzed.**
**At the end of the fourth paragraph of subsection 3.1, we added the sentences "Zn and Pb are two common anthropogenic trace elements in urban air. Their ratios to Fe in the dust samples in the postfrontal air were much lower than those in the prefrontal air and very close to those in the desert air (Table 3), further suggesting the limited influence of pollution on desert dust particles in the postfrontal air."**

*Typo: there is no April 31st.*
**Response: It is April 30. We corrected in the revised version.**

*Lines 12-13, P5: Even though I have not seen the supplementary figures, I would in principal disagree that the changes in dust particles during transport would be the same for each event. That would need to be shown.*
**Response:** In the supplements, we show the back-trajectory routes from the desert site and the Xi'an site during two dust storm periods (Figure S2 and Figure S3) and also the vertical thermodynamic structure of postfrontal dust plumes (Figure S6) when the dust samples were collected. The figures show that the two dust storms were really very similar according to their transporting routes and thermodynamic structures. Since these data are from public sites and other simulations, we do not think that it is a good idea to show them in the main body of the manuscript.

Yes, it is not absolutely correct that "the changes in dust particles during transport

would be the same for each event", and every dust storm must be more or less different from other dust storms. However, this does not mean we cannot find new understandings from a single dust storm which are common for dust storms. A single dust storm should have some common characteristics in a number of dust storms from the same desert. For your reference, we show the figures here to illustrate the similarities of the transport and the vertical thermodynamic structures of the two dust storms from which we collected samples.

[Figure]

Figure S2: Backward trajectories from the desert site (2014/04/24) and Xi'an site (2014/05/01) from the HYSPLIT model (www.arl.noaa.gov/HYSPLIT.php). (BST = GMT + 08:00)

[Figure]

Figure S3: CFORS model output for boundary layer (surface - 1000m) dust concentration ($\mu g/m^3$, color in log scale) and wind vector at 1000m of East Asia during the sampling periods at desert site (a) and Xi'an (b). (http://www-cfors.nies.go.jp/~cfors/index-j.html) (JST = GMT + 09:00)

[Figure]

Figure S6: Vertical profiles of virtual potential temperature near the surface at Yinchuan (38.48°N, 106.21°E), the WMO sounding station closest to the desert site, and at Jinhe (34.43°N, 108.97°E), a suburb place of Xi'an, before and after dust occurrence at the two places. The profiles were from the homepage of Atmospheric Soundings of the University of Wyoming (http://weather.uwyo.edu/upperair/sounding.html). Dust occurred at the desert site on the morning of April 24, 2014, and the sample collection was held between 06:30 and 15:00 BST on April 24. Dust occurred at Xi'an site on the morning of May 1, 2014, and the sample collection was held between 07:00 and 19:00 BST on May 1.

***Lines 20-25, page 7: This is one of the fundamental problems with trying to interpret this data. They have no way to distinguish between sulfate from pollution and sulfate in the soil/dust itself.***

**Response:** Yes, it could be absolutely said that there is "no way to distinguish between sulfate from pollution and sulfate in the soil/dust itself". However, what we are discussing here is if the small level of sulfate observed at the urban site (0.9%) was considerably larger than the level at the desert site (1.2%) and if the production of sulfate by surface reactions on dust particles during the particle travel was substantially large and has to be considered. Even the 0.9% of sulfate was totally from anthropogenic pollution, this does not contradict our conclusion that the production was limited. By the way, we do not have a reason to ignore the part of sulfate of mineral origin (1.2% in the present study) in the dust particles.

***Page 9, lines 19-21: "...very different from the conclusions of this study." What evidence is there that this study's "enhanced" (for purposes of discussion) nitrate was collected in transit vs from the populated area near the sampler? I believe this study's Conclusions are unsupported.***

**Response:** Below-detection-limit ammonia and unenriched Zn and Pb (data added in the revised manuscript) relative to mineral dust in the postfrontal air indicate that the nitrate was impossibly explained by possible emissions from the populated area near the sampler. So we consider the nitrate was produced during the transport, although the amount was very limited in comparison with that in polluted urban atmosphere. Even though some of the "enhanced" nitrate was from the populated area near the sampler, the production of nitrate should be very

small in comparison with that in polluted air, which supports our conclusion. Please also see our response to your next comment. In addition, it is very hard for us to believe the limited nitrate was produced in the last moment only before we collected the particles, because the conversion of background-like nitric acid to particle surface in dust air during the transport according to our estimation can, to a large extent, account for the nitrate production.

*Page 9, line 30-31: Yes, prefrontal air is much more polluted than postfrontal air. But that doesn't prove that the postfrontal air is free of contamination. The postfrontal air is still moving across a landscape containing sources, especially near the sampling site in Xian. How rapidly would urban nitrate be formed, relative to the sampling interval in the postfrontal air?*

**Response:** We didn't attempt to show "that the postfrontal air is free of contamination" and we never say that in the manuscript. We show that the production of sulfate and nitrate on dust storm particles were limited. With our data, we estimated the rate of nitrate formation in the postfrontal dust when the particles travelled from the desert area to the urban area, as we show in the manuscript. The adiabatic state of the postfrontal dust plume was kept during the travel. So the rate was very small. Although there should be emissions of anthropogenic pollutants from local areas where the plume passed, the emitted amount was not large enough to influence the dust plume. Otherwise, gaseous pollutants such as $SO_2$ and $NO_2$ would not have decreased to very low levels. The major reason should be that the movement of the postfrontal air was relatively very fast, in comparison with prefrontal air.

*Furthermore, since there was only bulk sampling we don't know for sure that all the nitrate was even on the coarse (dust) mode. Their observations are simply too few and too limited in type to advance our understanding of the uptake of sulfate and nitrate by desert dust.*

**Response:** Currently, we do not have size-differentiated data for the formation of the salts to give a deeper discussion on in what size ranges of dust particles the salts were produced. However, as we mentioned in previous responses, even we consider all nitrate and sulfate were produced on the dust storm particles, the production of the salts was still very limited, which does not contradict our conclusions. Yes, it is true our methods are not advanced and the observations did not provide a lot of samples. Repeatedly, we think that the key point should be if our results support our conclusions, no matter whether the methods are advanced or not, and how many samples we have. To the extent of our knowledge, we did not find data that contradict our data and conclusions. The differences between our data and published data were reasonably explained.

*Page 10, line 4: Briefly explain "Peak 1" or don't mention it.*

**Response:** Peak 1 in the study of Zhao et al. (2007) referred to the period of the highest loading of mass during the dust period.

**In the revision, we removed "Peak 1" from the text.** The sentence **"…, the mineral/TSP ratios in samples with the highest TSP loading (Peak I described in that study) were significantly lower than those in samples collected after the occurrence of maximum aerosol loading, indicating that the samples around Peak I were not dust particles from desert areas only."** was changed to **"…, the mineral/TSP ratios in samples**

**of the highest TSP loading were significantly lower than those in samples collected after the occurrence of maximum aerosol loading, indicating that the samples at the highest TSP moment were not dust particles from desert areas only."**

*I really like most of the discussion on page 10, which addresses a way of identifying urban vs desert influences on dust composition using trace metals. Unfortunately, this study only measured Ca, which is present in both desert and urban dust, so their conclusions can't benefit from this discussion.*

**Response:** In this section, we discussed the indictors of discriminating desert dust from urban aerosols in some previous studies to explain why some studies encountered the result that some "dust samples" contained substantial sulfate and nitrate. We emphasize that $Ca^{2+}$ is present in both desert dust and urban mineral particles and is not as a good indicator for discriminating desert dust from urban aerosols. Dust samples in studies using $Ca^{2+}$ as the indicator of the presence of desert dust could be a mixture of long-distance transported dust particles and locally- and regionally-originated aerosols, which caused the conclusion in some previously published papers that dust particles significantly enhanced the formation of sulfate and nitrate when dust plumes advected over urban areas. So the discussion helps to elucidate the discrepancy between the results of different studies as mentioned in the Introduction.

In our study, we did not use metals as indicators to discriminating desert dust from urban aerosol. We divided the sampling periods into three stages: prefrontal, frontal and postfrontal air. We found that the production of nitrate and sulfate in samples dominated by desert dust particles (in the postfrontal air) was very limited and we explained the results based on the adiabatic movement of the postfrontal dust plume.

In addition, we also measured other metals. In the revised manuscript, we will add the results of two common anthropogenic trace elements Zn and Pb, as mentioned in previous responses. The ratios of them to Fe in postfrontal dust particles were very close to those in the desert air, and much smaller than those in the prefrontal air (Table R2), further suggesting the limited influence of pollution on desert dust particles.

*Reviewer# 2*
**The authors greatly appreciate the careful and detailed review by the reviewer 2**
* * *
*General comments:*

A number of laboratory and field studies have proved that Asian dust particles readily promoted the formation of sulfate and nitrate when the lofted dust plumes transported across urban areas under high RH and elevated levels of reactive trace gases (i.e. $SO_2$, $NOx$, $O_3$ and $OH$ radicals). This would significantly alter the physical and chemical properties of dust aerosol and subsequent climate change on regional scale.

The authors carried out a series of particle samplings at the Tengger desert (06:30–15:00 BST on April 24, 2014) and downwind Xi'an city (07:00–19:00 BST on May 1, 2014) during two independent dust storms. Combination of HYSPLIT backward trajectories model and CFORS simulation, they showed that the two dust events originated from the same source regions and had similar transport routes. They compared the concentrations and mass fraction of chemical components (i.e. sulfate, nitrate, ammonia, and elemental ratios) in dust particles at two sites during the prefrontal, frontal, and postfrontal air parcels, and indicated that the production of sulfate and nitrate on front-associated dust particles was limited when the dust moved from desert sources to populated areas in northwest China. The result of this manuscript seems to be reasonable in spite of limited in-situ sampling data, which is completely different from the other previous studies. Different scientific viewpoints should be encouraged to promote the understanding of interplay between mineral dust and atmospheric chemistry. Therefore, I recommend this manuscript is accepted and published in the journal of ACP after some revisions.

*Specific comments:*

*1.* Abstract, Page 2, lines 12–14: "The significant sulfate and nitrate reported in dust-associated samples in previous studies were more likely produced on locally-emitted and urban mineral particles or from soil-derived sulfate rather than being produced via chemical conversions on desert dust particles."

Conclusion, Page 11, lines 2–4: "Significant sulfate and nitrate in dust storm periods in China reported in previous studies were likely produced on locally-emitted and urban mineral particles, in addition to soil-derived sulfate, and they were unlikely produced via chemical conversions on dust particles from deserts."

**Comment:** *I think there is no enough evidence for the manuscript to demonstrate this conclusion. Because the dry condition (RH<40%) and low mass concentrations of trace gases (i.e. $SO_2$ and $NO_2$) were observed in Xi'an during the dust-storm episode, which didn't favor the formation of sulfate and nitrate on the surface of mineral particles. However, these are only a few cases. The authors didn't show the results when dust storm transported across the other polluted areas with high RH conditions and high levels of trace gases.*

**Responses**: The mentioned conclusion is not derived from our data. It is from the careful investigation of the sample collection records in published literature where significant sulfate and nitrate in so-called dust samples (Section 3.3) were reported. We checked all papers that we can find on the formation of sulfate and nitrate on Asian dust particles by field measurement data. Unfortunately, most of the papers did not give the details of the dust conditions and the evolution of

weather conditions at the start and stop time of sample collection. To the extent that we can confirm the dust and weather conditions at the start and stop time of samples collection, results from the samples that were really collected under dust storm conditions are all similar to the results we reported in this study, as described in Section 3.3.

We do not intend to show that dust storm particles cannot enhance the production of sulfate and nitrate via chemical conversions on their surface. We want to report that dust storm particles did not enhance the production in postfrontal air within the Asian continent. The reason is the adiabatic state of the air parcels loading the dust particles, which is the reason for the low RH and less $SO_2$ and NOx. We totally agree that the absence of the sulfate and nitrate in the samples of dust storm particles of ours were due to the dry conditions and the low concentration of $SO_2$ and NOx. In case when dust particles are put into an environment with high RH and high concentrations $SO_2$ and NOx, sulfate and nitrate can be produced efficiently. However, this is not the case of long-distance transport of dust storm plumes following cold front within the Asian continent. In cases of cyclone with cold fronts, the postfrontal air is always dry and its arrival is always accompanied with a rapid decrease of anthropogenic pollutants including $SO_2$ and NOx. There are a lot of papers on this point for dust arrival in East China and we cited some, such as Hu et al (2016), to mention this. Please see section 3.3.

We prefer to remain the explanation for the difference of our results with some previous studies, although we do not have evidence from our data. The reasons are: (1) to the extent of our knowledge, we can give such an instant explanation; (2) the explanation is referentially meaningful for further studies of the formation of nitrate and sulfate on dust particles being transported within the Asian continent in postfrontal air; and (3) if we do not provide a possible explanation, readers will have a question "how do the authors explain/consider the previously-published results?" after reading the abstract.

To avoid the misunderstanding, we made some revisions in the Abstract and Conclusion:
In the abstract,
**"The significant sulfate and nitrate reported in dust-associated samples in previous studies were more likely produced on locally-emitted and urban mineral particles or from soil-derived sulfate rather than being produced via chemical conversions on desert dust particles."**
is changed to
**"The significant sulfate and nitrate reported in dust storm-associated samples in previous studies were more likely from locally-emitted and urban mineral particles or from soil-derived sulfate, because the weather conditions in those studies indicated that the air from which the samples were collected very likely contained a lot of particles from local emissions."**

In the revision of Conclusion, in addition to make the above point more clear, we also mention the need of an effort to quantify the contribution of non-desert mineral particles to the sulfate and nitrate in future studies.
**In the revision, the description of "Significant sulfate and nitrate in dust storm periods in China reported in previous studies were likely produced on locally-emitted and urban mineral particles, in addition to soil-derived sulfate, and they were unlikely produced via chemical conversions on dust particles from deserts."**
is changed into (as an individual paragraph)

**"Significant sulfate and nitrate in dust storm periods in China reported in previous studies were likely from locally-emitted and urban mineral particles, in addition to soil-derived sulfate, and they were unlikely produced via chemical conversions on dust particles from deserts. The major reason is that, in those studies, the air from which the samples were collected had been significantly influenced by local emissions. Without a proper evaluation of the contribution of sulfate and nitrate in the samples by locally-emitted and urban mineral particles, i.e., non-desert mineral particles, it is not safe to attribute all the detected sulfate and nitrate to the production on dust-storm particles."**

*2. Page 4, lines 21–22: "This sample collection ensured that mineral particles collected on the filters were dust particles from the desert and there should be no influence of anthropogenic pollutants from the village or the city considered in the samples."*
**Comment:** *The evidence provided by this manuscript could not fully support this sentence. Please reconsider again.*
**Response:** As we described in the manuscript, we collected the samples carefully with a proper time resolution for describing the variation with weather change. We chose the samples as dust storm samples when the air was from the desert and the same direction. We consider that possible influence from the village and the city was avoided when the wind was from the desert direction.
In the revision, the descriptions were revised into **"This sample collection ensured that mineral particles collected on the filters were dominated by dust particles from the desert and possible influence of anthropogenic pollutants from the village or the city was suppressed."**

*3. Page 6, lines 15–17: "The cold fronts are the boundaries between the local or regional anthropogenic-polluted air and the long-distance transported air because the movement of air on a synoptic scale is approximately adiabatic, i.e. the air is hardly mixed with thermodynamically-different air it meets."*
**Comment:** *I don't agree with this viewpoint about "the air is hardly mixed with thermodynamically-different air it meets". In terms of meteorology, the warm and humid air mass is readily lifted and the weather process (e.g., strong wind and cooling weather, rainfall or snow) often changes dramatically on the border of frontal system when a cold front passes over. As shown in Figure 2a, the RH increased sharply from 40% at 13:00 BST to 100% at 16:00 BST, which indicated clearly that a rainfall or snow process took place at Tengger desert (also see Page 4, line 15 and Table 1). The cold fronts are dominated and accompanied strong winds intensify the diffusions of local air pollutants.*

**Response:** We totally agree that "the weather process (e.g., strong wind and cooling weather, rainfall or snow) often changes dramatically on the border of frontal system when a cold front passes over." However, this occurs only in the front. In the case of the presence of a cold front, the cold and dry postfrontal air did not mix with the prefrontal air in the viewpoint of air movement on synoptic scales, which is the reason of the presence of the front. We also totally agree that "the cold fronts are dominated and accompanied strong winds intensify the diffusions of local air pollutants." However, the major part of accumulated pollutants in prefrontal air is usually pushed by the front, move northeastward and separately from postfrontal dust storm plumes, and transported out of the Asian continent. Only a small part of the pollutants close to the front might be mixed with dustloading air in the front. In addition, this is very different from dust particles in marine atmosphere. In our previous studies in the downwind marine atmosphere of the Asian continent (in southwestern Japan), we have confirmed the mixture, that is likely due to the vertical mixing in the marine atmosphere between China coast and Japanese islands.

On the RH=100% at 16:00 BST in Figure 2a, the dust storm has finished at that time and the weather was recovering with the increase of temperature and RH. The high RH was caused by the arrival of another air parcel which was dryer and much colder than the previous air, leading to snowing.

In the revision, **"although some small-scale mixing might occur in the front"** is added after the mentioned sentence.

*4. Page 7, lines 26–33: "At the desert site, $NO_3^-$ concentration in dust samples was 4-6 $\mu g\ m^{-3}$ and the average was 5 $\mu g m^{-3}$. The relative amount of $NO_3^-$ range between 0.11% and 0.12%, and the average was 0.12%. ...Right after the passage of the cold front (the first sample in the postfrontal air), the concentration of $NO_3^-$ was 0.9 $\mu g m^{-3}$ and it occupied 0.2% of the aerosol mass. The relative amount in this sample was about twice of that in the desert samples although it was the lowest in the samples at Xi'an site, indicating that nitrate had been produced on dust particles during their travel to Xi'an."*

***Comment:*** *At the Tengger desert site, $NO_3^-$ concentration in dust samples was 4-6 $\mu g\ m^{-3}$ (with the average value and fraction of 5 $\mu g\ m^{-3}$ and 0.12%), which were much larger than that at Xi'an after the cold front (~0.9 $\mu g m^{-3}$, with the mass fraction ~0.2%). The higher mass fraction of $NO_3^-$ at Xi'an was ascribed to the low concentration of TSP (total suspended particulates, ~420 $\mu g\ m^{-3}$), and the TSP concentration in Tengger desert site was about 5000 $\mu g\ m^{-3}$. Although the relative amount of $NO_3^-$ at Xi'an was about twice of that in the desert samples, it couldn't indicate that nitrate had been produced on dust particles during their travel to Xi'an. Please explain this.*

**Response:** Yes, we agree that "Although the relative amount of $NO_3^-$ at Xi'an was about twice of that in the desert samples, it couldn't indicate that nitrate had been produced on dust particles during their travel to Xi'an". A possibility we did not consider is that the increase was caused by possible difference of removal rate of dust particles and nitrate (or nitrate-containing particles). If this possibility were the fact, it means that part of the nitrate, similar to sulfate, was not produced in the dust-loading plumes, or the production rate was smaller than our estimation, both of which do not conflict with our conclusion that the production of nitrate was limited.

**In the revision,**
**(1) "indicating that nitrate had been produced on dust particles during their travel to Xi'an"** was revised into **"indicating that nitrate was likely produced on dust particles during their travel to Xi'an."**
(2) The following descriptions are added right after the descriptions of the estimated rate values of nitrate production on dust particles: **"Note this rate should be the maximum rate because not all the nitrate must have been produced on dust particles and the increase of the relative amount of nitrate during the movement of a dust plume from the desert to Xi'an could have been the consequence of possible difference of removal rates of dust particles and nitrate-containing particles."**

**5.** *Page 18, Table 2; Page 19, Table 3: The authors sampled the concentrations of TSP (total suspended particulates) and analyzed the chemical components (i.e. sulfate, nitrate, and ammonia) in TSP at the Tengger desert and Xi'an sites.*

**Comment:** *Please explain why did you sample the concentrations of TSP, instead of PM$_{10}$ or PM$_{2.5}$. It is well known that most of the coarse-size dust particles (radii > 10 μm) generally settle near the source region on account of large gravitational deposition velocity, whereas the finer dust particles (radii < 10 μm) are transported more efficiently to the downstream areas. And the concentrations of TSP (meaning coarse-size particle with radii>10 μm) in Xi'an city should include the local source emissions (e.g., engines of vehicles, road dust, and construction dust; Page 4, lines 27-29) that increases the TSP concentrations, but may decrease the relative mass fraction of sulfate, nitrate, and ammonia. Inferred from Page 6, Lines 1-7, the mass concentrations of TSP and sulfate are about 425 μg m$^{-3}$ and 17 μg m$^{-3}$ at Xi'an before the dust arrival (i.e. prefrontal air). In the postfrontal air, the mass concentrations of sulfate are 3.8, 3.5, and 3.4 μg m$^{-3}$, respectively, right after, two hour after, and four hours after the passage of cold front (means slight variations), whereas the corresponding TSP concentrations are 422, 318, and 189 μg m$^{-3}$ (shows large variations). Clearly, relative mass fractions of sulfate reduce.*

**Response:** Size-differentiated sulfate and nitrate can give a deeper understanding on the variation of the sulfate and nitrate on dust particles. Samples of different size fractions are always collected using Anderson sampler. Unfortunately, the sampling site in the desert was located at an active sand dune and we did not have an electricity power to support such sample collections. Moreover, it is difficult to collect enough samples of different size fractions for water-soluble ions analysis using the Anderson sampler within 1-2 hours. So we collected TSP (total suspended particulates) samples. It is possible that the content of mineral sulfate in dust particles is a size matter and the size dependence must have large influence on the soil-derived sulfate in downstream areas, which is an important subject in future studies.

   At Xi'an, the relative mass fractions of sulfate increased gradually with the decrease (the leaving of the front) of TSP. This result is consistent with previous studies demonstrating the passage of cold fronts, including on-line instrument measurements such as Wang et al. (2014) at Xi'an, and Niu et al (2016) and Hu et al. (2016) at Beijing. It is because the gradual increase of the influence of local emissions as the front leaves away. Anyway, the concentration of sulfate in postfrontal dust air was very low, in comparison with usual urban polluted air. Even we consider part of the sulfate we encountered at Xi'an was produced on desert dust particles via surface reactions, the production was still very low, which is consistent with the major conclusion of this study.

Hu, W., Niu, H., Zhang, D., Wu, Z., Chen, C., Wu, Y., Shang, D. and Hu, M.: Insights into a dust event transported through Beijing in spring 2012: Morphology, chemical composition and impact on surface aerosols, Sci. Total Environ., 565, 287–298, doi:10.1016/j.scitotenv.2016.04.175, 2016.

Niu, H., Hu, W., Zhang, D., Wu, Z., Guo, S., Pian, W., Cheng, W., Hu, M., 2016. Variations of fine particle physiochemical properties during a heavy haze episode in the winter of Beijing. Sci. Total Environ. 571, 103–109. doi:10.1016/j.scitotenv.2016.07.147

Wang, G. H., Cheng, C. L., Huang, Y., Tao, J., Ren, Y. Q., Wu, F., Meng, J. J., Li, J. J., Cheng, Y. T.,

Cao, J. J., Liu, S. X., Zhang, T., Zhang, R. and Chen, Y. B.: Evolution of aerosol chemistry in Xi'an, inland China, during the dust storm period of 2013—Part 1: Sources, chemical forms and formation mechanisms of nitrate and sulfate, Atmos. Chem. Phys., 14(21), 11571–11585, doi:10.5194/acp-14-11571-2014, 2014.

**Minor comments:**

*1. Abstract, Page 2, lines 3–4: "but the production was very inefficient in other studies."*
**Comment:** *Please give the quoted literature.*
**Response:** The literature is given in the second paragraph of the Introduction. Since this part is abstract, we prefer not to list references here.

*2. Page 3, line 8: "RH"*
**Comment:** *Change to "relative humidity (RH)"*
**Response:** Corrected in the revision.

*3. Page 10, line 4: "mineral/TSP ratios"*
**Comment:** *Change to "mineral/TSP (total suspended particulates) ratios"*
**Response:** Corrected in the revision.

[revised manuscript text omitted]

---

## Author Response (AR2)

June 12, 2017

Dear editor

Here, we submit the revised manuscript (acp-2016-853) entitled, "Limited production of sulfate and nitrate on front-associated dust storm particles moving from desert to distant populated areas in northwestern China" to the special issue of "Anthropogenic dust and its climate impact" of your journal *Atmospheric Chemistry and Physics*. We would like to thank you for your constructive comments.

Attached are the list of changes made in the manuscript, our point-by-point responses to your comments and a marked-up manuscript version in which all changes were marked with underlines.

Thanks for your help.

Yours sincerely,

Daizhou Zhang

\*\*\*\*\*\*\*\*\*\*\*\*\*\*\*\*\*\*\*\*\*\*\*\*\*\*\*\*\*\*\*\*\*\*\*\*\*\*\*\*\*\*\*\*\*\*\*\*\*\*\*\*\*\*\*\*\*\*\*\*
Faculty of Environmental and Symbiotic Sciences

Prefectural University of Kumamoto

Kumamoto 862-8502, Japan

Tel:+81-(0)96-321-6712   Fax:+81-(0)96-384-6765

Email: dzzhang@pu-kumamoto.ac.jp
\*\*\*\*\*\*\*\*\*\*\*\*\*\*\*\*\*\*\*\*\*\*\*\*\*\*\*\*\*\*\*\*\*\*\*\*\*\*\*\*\*\*\*\*\*\*\*\*\*\*\*\*\*\*\*\*\*\*\*\*

**List of changes made in the manuscript**

1. We added the citations of **Bi et al. (2011)** and **Fu et al. (2009) in line 3 of Page 3, Wang et al. (2012) in line 29 of Page 3**.
2. We added the bibliography of **Bi et al. (2011) in line 14-16 of Page 12, Fu et al. (2009) in line 4-5 of Page 13**, and **Wang et al. (2012) in line 28-30 of Page15.**

These references are added in the reference list as:

Bi, J., Huang, J., Fu, Q., Wang, X., Shi, J., Zhang, W., Huang, Z., Zhang, B.: Toward characterization of the aerosol optical properties over Loess Plateau of Northwestern China. J. Quant. Spectrosc. Radiat. Transf. 112, 346–360. doi:10.1016/j.jqsrt.2010.09.006, 2011.

Fu, Q., Thorsen, T.J., Su, J., Ge, J.M., Huang, J.P.: Test of Mie-based single-scattering properties of non-spherical dust aerosols in radiative flux calculations. J. Quant. Spectrosc. Radiat. Transf. 110, 1640–1653. doi:10.1016/j.jqsrt.2009.03.010, 2009.

Wang, J., Xu, X., Henze, D.K., Zeng, J., Ji, Q., Tsay, S.-C., Huang, J.: Top-down estimate of dust emissions through integration of MODIS and MISR aerosol retrievals with the GEOS-Chem adjoint model. Geophys. Res. Lett. 39, L08802, doi:10.1029/2012GL051136, 2012.

**Point-by-point response to the comments from the editor**

**The authors greatly appreciate the comments from the editor**
* * *
*Comments to the Author:*

*You have revised the manuscript following the comments and suggestions and significantly improved the manuscript. Therefore, I decide to accept for publication. However, you should check the reference one by one and make sure they are right.*

**Reply:** We checked the references and their citation one by one and made sure that all references are cited correct. Three references were thought necessary and included into the revised manuscript.

In addition, we would like to add the citation of another three references in this revision: **Bi et al. (2011)** and **Fu et al. (2009) in line 3 of Page 3, Wang et al. (2012) in line 29 of Page 3**. The bibliography of **Bi et al. (2011) was included in line 14-16 of Page 12, Fu et al. (2009) in line 4-5 of Page 13**, and **Wang et al. (2012) in line 28-30 of Page15.**

[revised manuscript text omitted]

---

## Author Response (AR3)

October 21, 2017

Dear editor

Here, we submit the revised manuscript (acp-2016-853) entitled, "Limited production of sulfate and nitrate on front-associated dust storm particles moving from desert to distant populated areas in northwestern China" to the special issue of "Anthropogenic dust and its climate impact" of your journal *Atmospheric Chemistry and Physics*. We would like to thank you for your constructive comments.

Attached are the list of changes made in the manuscript, our point-by-point responses to your comments and a marked-up manuscript version in which all changes were marked with underlines.

Thanks for your help.

Yours sincerely,

Daizhou Zhang
* * *
Faculty of Environmental and Symbiotic Sciences
Prefectural University of Kumamoto
Kumamoto 862-8502, Japan
Tel:+81-(0)96-321-6712   Fax:+81-(0)96-384-6765

Email: dzzhang@pu-kumamoto.ac.jp
* * *
**List of changes made in the manuscript**

1. **In line 9 on page 2**, we replaced "indicate" with "suggest" in the revision.
2. **At the beginning of the last sentence of the abstract (line 12-13 on page 2)**, "To the extent of our investigation of published literature," is added.
3. **In line 14 on page 2** "dust" was inserted before "storm-associated".
4. **At the end of the abstract (line 16-18 on page 2)**, "Therefore, for an accurate quantification of sulfate and nitrate formed on long-distance transported desert dust particles at downwind populated areas in eastern China, efforts are indispensable in dust collection to minimize any possible influence by locally-emitted particles or, at least, to ensure that the samples are collected after dust arrival." is added.
5. **In line 27-28 on page 3**, The sentence "what are the reasons for the discrepancy between the results of those studies?" was changed to "why are there so much different rates of sulfate and nitrate production during dust transport to polluted areas?"
6. **In line 1-2 on page 4**, The sentence "and discuss possible reasons for the difference in previously reported results" was changed to "to understand the chemistry/aging on dust"
7. **In line 18-20 on page 4**, the sentence "Right after the sample collection of each filter, the filter was put into a polystyrene petri dish, which was in turn sealed in a plastic bag and stored into refrigerators at -1°C until subsequent analyses" is added.
8. **In line 9-10 on page 5**, the sentence "Each sample filter was put into a polystyrene petri dish, which was in turn sealed in a plastic bag and stored in a refrigerator at -1°C until subsequent analyses" is added.
9. **In line 33 on page 5 – line 1-2 on page 6**, "Samples and blanks collected at both sites were analyzed with replicates and surrogates following standard lab protocols. The relative uncertainties in the mass percentage of sulfate and nitrate, according to the surrogates, were less than 10%." is added.
10. **At the end of Section 3.3 (21-34 on page 11 – line 1-2 on page 12)**, the following paragraph is added.

    To further examine the situation of previously-reported sulfate and nitrate formation on dust-storm particles at populated areas in eastern China, we investigated all published papers we found on this subject, and carefully checked the records of sample collection and the available meteorological conditions when the samples were collected in those studies, from the papers and officially public web sites for historical meteorological records. The papers were separated into three groups according to the records and meteorological conditions, as mentioned above. The first group includes papers in which the sample collection records were vague, and we are unable to make clear if the samples were dominated by desert dust particles or contained a large fraction of locally-emitted particles (Table S2a). It is not sure that the sulfate and nitrate reported in those papers were really on desert dust particles or not. The second group includes papers in which the sample collection was started before the arrival of the fronts of dust-loading cyclones or the fronts had disappeared and front-associated dust-loading air had mixed with locally-polluted air (Table S2b). That means the samples in the studies contained not only long-distance transported desert dust particles but also locally-emitted mineral particles, such as road dust, construction dust and fly ashes. In such samples, the sulfate and nitrate must have been substantially contributed by locally-emitted mineral particles, as we discussed above. The third group includes papers in which results from samples of locally- or regionally-originated particles in prefrontal air and from samples of long-distance transported desert dust particles in postfrontal air can be identified (Table S2c). The production of sulfate and nitrate in the postfrontal dust particle samples was all very limited, and the production in prefrontal samples was significant, which is consistent with our results in this study.

11. **Line 10-11 on page 12,** the first sentence of the second paragraph was modified into "Significant sulfate and nitrate in dust storm periods in China reported in previous studies in reality for the most part probably did not link to reactions on the dust surface. They were likely from locally-emitted and urban mineral particles, in addition to soil-derived sulfate."

12. **At the end of the Section 4 Conclusion (16-27 on page 12)**, the following paragraph is added.

The results of this study are from the comparison of dust particles in two dust storms: one at the dust source area, and another at an urban area after long-distance transport. Although the thermodynamic structure of the dust-loading air in the two cases was similar and comparative, data from multiple cases of same dust storms at desert areas and downwind populated areas are needed to make the conclusions more accurate and confident. In addition, the conclusions were derived for front-associated dust storm particles. The adiabatic nature of the postfrontal air during its long-distance movement kept the air dry and hardly polluted by accumulated anthropogenic pollutants from areas where it passed. There are other types of airborne dust particles in China, such as floating dust. The movement of the air loading floating dust is usually slow and not adiabatic, and the air is usually well mixed with locally-emitted pollutants, which is very different from postfrontal air. It can be expected that floating dust particles could be more frequently changed to sulfate- and nitrate-carriers via surface chemical reactions than front-associated dust storm particles in urban areas. However, how to separate the sulfate and nitrate produced on floating dust particles from those produced on locally-emitted mineral particles is still a big challenge in field observations, because floating dust particles and locally-emitted mineral particles coexist in urban air when floating dust occurs.

13. **In the supplementary information**, Table S2a-c and relative references were added.

**Point-by-point responses to reviewer comments:**

**Major comments:**

*The paper by Wu et al. analyses desert dust aerosol composition at two sites, one close the source in Asia and the other one in an urban area after 700 km transport, and investigate the possible formation of nitrates and sulfates due to particle chemical aging during transport. The main conclusion of the paper is that very small quantities of sulfates and nitrates are observed after transport compared to dust close to their lifting source. The authors argue that sulfate and nitrate production reported by previous studies for dust are more likely due to local production on urban mineral particles or soil derived components that chemical reactions on dust. Based on past literature (both field and laboratory studies) and the careful analysis of the methodology and results presented in this paper, I feel to recommend the paper to be considered for publication on ACP only after substantial revisions. These revisions to me mostly concern the presentation and discussion of the data and the conclusions. In fact, I mostly agree with one of the two reviewers in stating that the conclusions drawn here are not really supported by the data, i.e. only one case of source and transported dust is analysed here and also not the same dust episode is investigated but two different ones. The authors answered this comment saying that it is not necessary to have too many observations to obtain a good result (and that the two dust cases analysed are comparable), and I agree with that, but at the same time I find the conclusions they gave in the paper too much conclusive considering the provided dataset. I would avoid generalizing the results and conclude that sulfate and nitrate observed in previous studies are in reality for the most part probably not linked to reactions on the dust surface. Many different field conditions have been found in the literature, i.e. temperature, relative humidity, or precursor gas concentration, and may explain the different findings. Probably the authors should go deeper to all past literature and compare all the conditions/results to be able to state more general ideas (indeed some useful discussion is already provided in Sect. 3). But also in this case the single observation provided in this study is not sufficient to me to contradict previous studies. It may provide new results that can be considered in conjuction with previous ones. For this reason I feel also to agree with the other of the two reviewers in considering this dataset as potentially very useful to the scientific community and for this reason worthy of being published. So, in conclusion I would encourage the authors to «smooth» their conclusions and to revise / exson of their data / results and to put them better in the literature context so to give the reader the appropriate elements to evaluate the significance of the provided dataset.*

**Reply:**

The authors greatly appreciate the careful review and helpful comments.

In this revision, to avoid the generalization of the results and conclusions and to smooth the conclusions, at the end of the abstract and the end of the conclusion section, descriptions are added. The major points of the added descriptions include (1) it is important to separate long-distance transported desert dust particles from locally-emitted mineral particles in sample collection at urban areas; (2) the conclusions of this study were derived from the comparison of two different dust-storm episodes and it is necessary to verify the conclusions with more data comparison of same dust-storm episodes; (3) as shown

in the paper title, the present conclusions are limited to front-associated dust storm particles; (4) there are other kinds of desert dust particles, such as floating dust, in China and further studies are necessary for floating dust.

To further show the situation of sulfate and nitrate formation on dust particles in past literature, in this revision, we add the information of the published papers on this subject we found as possible as we could. The papers were separated into three groups according to whether the samples in the studies were mainly composed of desert dust particles or were mixture of locally-emitted particles and desert dust particles, or the distinctiveness could not be confirmed from the records reported in the papers. The relevant results and discussion are added as an individual paragraph at the end of Section 3.3. Three tables, (Tables S2a, S2b, S2c) showing the papers, the major points related to sulfate and/or nitrate formation in the papers, relevant information of the samples used in the papers, and our verification, are added into the supplementary materials.

In accordance, following 4 revisions are made for the major comments in the revised manuscript.

[revised manuscript text omitted]

**Specific comments**

*I will give just few specific comments since a large part of my comments/questions have been already raised by the other two reviewers.*

*Abstract, line 9-10 (and generally in the text): I would replace the verb « indicate » with « suggest » or similar when you state on some conclusive statement which could be not really proven*

**Reply:** We replaced "indicate" with "suggest" in the revision.

*Introduction, page 3, line 26: to me it is not really a matter of different results but probably of different conditions leading to different reactions on dust. I would to me the question is: why there are so much different rates of production of sulfate and nitrate during dust transport to polluted areas? And what you bring with your study is another observation to add to the piece of work already done in the literature to try to understand the chemistry/aging on dust.*

**Reply:**

The sentence "what are the reasons for the discrepancy between the results of those studies?" was changed to "why are there so much different rates of sulfate and nitrate production during dust transport to polluted areas?"

The sentence "and discuss possible reasons for the difference in previously reported results" was changed to "to understand the chemistry/aging on dust"

*Section 2: based on your sampling and analysis protocol I wonder if you do not risk to lose your sulfate and nitrate before analysis. Supposing that these are liquid coatings on dust, are you sure that they do not evaporate from dust before analysis? This is a key point of the methodology that should be discussed in the paper.*

**Reply:** The possibility that dust particles had a liquid coating was very small because the air was very dry when the dust samples were collected. We had made efforts to suppress any possible loss of sulfate and nitrate on the surface of dust particles before analysis. We followed the standard operating program (SOP) of field sampling and transportation. After the sample collection of each filter, the filter was put into a polystyrene petri dish and then the dish was sealed in a plastic bag and stored in a car-carried refrigerator at -1°C at the desert site. After being taken back to the lab at Xi'an, the bags were transferred into a lab refrigerator. The samples at Xi'an were dealt in the same way and saved in the lab refrigerator at -1°C.

In the revision, "Right after the sample collection of each filter, the filter was put into a polystyrene petri dish, which was in turn sealed in a plastic bag and stored into refrigerators at -1°C until subsequent analyses" is added in **line 18-20 on page 4**.

"Each sample filter was put into a polystyrene petri dish, which was in turn sealed in a plastic bag and stored in a refrigerator at -1°C until subsequent analyses" is added in **line 9-10 on page 5**.

*Section 3: which is the uncertainty on the mass percent of sulfate and nitrate you obtain? The values at the source and transport region are mostly the same considering the uncertainties? This is another key element to discuss.*

**Reply**: In this study, the filters were analyzed following the standard lab protocols of ion analysis with IC and crustal analysis with ED-XRF. The analyses were run with replicates and surrogates. The uncertainties in the mass percentage of sulfate and nitrate were from the error propagation of the precision of replicate measurement of sulfate and nitrate in filter samples including blanks. The relative uncertainties 
[revised manuscript text omitted]

---

## Author Response (AR4)

November 2, 2017

Dear editor

Here, we submit the revised manuscript (acp-2016-853) entitled, "Limited production of sulfate and nitrate on front-associated dust storm particles moving from desert to distant populated areas in northwestern China" to the special issue of "Anthropogenic dust and its climate impact" of your journal *Atmospheric Chemistry and Physics*. We would like to thank you for your constructive comments.

Attached are the list of changes made in the manuscript, our point-by-point responses to your comments and a marked-up manuscript version in which all changes were marked with underlines.

Thanks for your help.

Yours sincerely,

Daizhou Zhang
* * *
Faculty of Environmental and Symbiotic Sciences

Prefectural University of Kumamoto

Kumamoto 862-8502, Japan

Tel:+81-(0)96-321-6712   Fax:+81-(0)96-384-6765

Email: dzzhang@pu-kumamoto.ac.jp
* * *
**List of changes made in the manuscript**

1. **Line 10 Page 2** in abstract of the last submission: "likely" was added before "limited", "within the continent" was added at the end of the sentence, and the whole sentence "These results suggest that the production of nitrate and sulfate on dust particles following cold fronts is likely limited when the particles move from the desert to populated areas within the continent." was moved to **Line 16 - 17 Page 2** in the revised manuscript.

2. **Line 12-15 Page 2** in abstract of the last submission: The sentence of "To the extent of our investigation of published literature, the significant sulfate and nitrate reported in dust storm-associated samples in previous studies were more likely from locally-emitted and urban mineral particles or from soil-derived sulfate, because the weather conditions in those studies indicated that the air from which the samples were collected very likely contained a lot of particles from local emissions." was revised into (**Line 11-16 Page 2** in the revised manuscript) "To the extent of our investigation of published literature, the significant sulfate and nitrate in dust storm-associated samples within the continental atmosphere reported in previous studies cannot be confirmed to be really produced on desert dust particles and the contribution from locally-emitted and urban mineral particles or from soil-derived sulfate was likely substantial, because the weather conditions in those studies indicated that the collection of the samples was started before dust arrival or the air from which the samples were collected was a mixture of desert dust and locally-emitted mineral particles."

3. **Line 16 Page 2** in the abstract: "Therefore," was removed.

4. **Line 5 Page 3** in the section of introduction: "the" was added before "uptake".

5. **Line 6 Page 3** in the section of introduction: "by" was added before "surface reactions".

6. **Line 19-20 Page 10** of last submission, Section 3.3 Inter-comparisons and implication: The sentence of "But the sulfate and nitrate were from the particles originating from the local and regional areas and they were not produced on dust particles from desert areas." was revised into (**Line 19-20 Page10** in the revised manuscript) "However, the sulfate and nitrate must have been contributed by particles in prefrontal air, which should be from local or regional areas and abundant in sulfate and nitrate."

7. **Line 19 Page 12** in the section of conclusion: "current" was replaced with "present".

**Point-by-point responses to editor's comments:**

*I would encourage the authors to « smooth » their conclusions and to revise / extend the discussion of their data / results and to put them better in the literature context so to give the reader the appropriate elements to evaluate the significance of the provided dataset.*

**Response**: The authors appreciate the helpful comments. To further smooth the conclusions and revise/extend the discussion, following three revisions were added in the revised manuscript.

1. **Line 10 Page 2** in abstract of the last submission: "likely" was added before "limited", "within the continent" was added at the end of the sentence, and the whole sentence

   "These results suggest that the production of nitrate and sulfate on dust particles following cold fronts is likely limited when the particles move from the desert to populated areas within the continent." was moved to **Line 16 - 17 Page 2** in the revised manuscript.

2. **Line 12-15 Page 2** in abstract of the last submission: The sentence of

   "To the extent of our investigation of published literature, the significant sulfate and nitrate reported in dust storm-associated samples in previous studies were more likely from locally-emitted and urban mineral particles or from soil-derived sulfate, because the weather conditions in those studies indicated that the air from which the samples were collected very likely contained a lot of particles from local emissions." was revised into (**Line 11-16 Page 2** in the revised manuscript)

   "To the extent of our investigation of published literature, the significant sulfate and nitrate in dust storm-associated samples within the continental atmosphere reported in previous studies cannot be confirmed to be really produced on desert dust particles and the contribution from locally-emitted and urban mineral particles or from soil-derived sulfate was likely substantial, because the weather conditions in those studies indicated that the collection of the samples was started before dust arrival or the air from which the samples were collected was a mixture of desert dust and locally-emitted mineral particles."

3. **Line 19-20 Page 10** of last submission, Section 3.3 Inter-comparisons and implication: The sentence of

   "But the sulfate and nitrate were from the particles originating from the local and regional areas and they were not produced on dust particles from desert areas." was revised into (**Line 19-20 Page10** in the revised manuscript)

   "However, the sulfate and nitrate must have been contributed by particles in prefrontal air, which should be from local or regional areas and abundant in sulfate and nitrate."

To polish the English *as requested in the decision letter*, the following corrections were made in the revised submission

4. **Line 16 Page 2** in the abstract: "Therefore," was removed.

5. **Line 5 Page 3** in the section of introduction: "the" was added before "uptake".

6. **Line 6 Page 3** in the section of introduction: "by" was added before "surface reactions".

7. **Line 19 Page 12** in the section of conclusion: "current" was replaced with "present".

[revised manuscript text omitted]